# Performance Evaluation of a Sensor Concept for Solving the Direct Kinematics Problem of General Planar 3-RPR Parallel Mechanisms by Using Solely the Linear Actuators' Orientations

**Stefan Schulz** 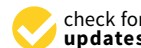

Workgroup on System Technologies and Engineering Design Methodology, Hamburg University of Technology, 21073 Hamburg, Germany; st.schulz@tuhh.de

**Abstract:** In this paper, we experimentally evaluate the performance of a sensor concept for solving the direct kinematics problem of a general planar 3-RPR parallel mechanism by using solely the linear actuators' orientations. At first, we review classical methods for solving the direct kinematics problem of parallel mechanisms and discuss their disadvantages on the example of the general planar 3-RPR parallel mechanism, a planar parallel robot with two translational and one rotational degrees of freedom, where P denotes active prismatic joints and R denotes passive revolute joints. In order to avoid these disadvantages, we present a sensor concept together with an analytical formulation for solving the direct kinematics problem of a general planar 3-RPR parallel mechanism where the number of possible assembly modes can be significantly reduced when the linear actuators' orientations are used instead of their lengths. By measuring the orientations of the linear actuators, provided, for example, by inertial measurement units, only two assembly modes exist. Finally, we investigate the accuracy of our direct kinematics solution under static as well as dynamic conditions by performing experiments on a specially designed prototype. We also investigate the solution formulation's amplification of measurement noise on the calculated pose and show that the Cramér-Rao lower bound can be used to estimate the lower bound of the expected variances for a specific pose based exclusively on the variances of the linear actuators' orientations.

**Keywords:** direct kinematics problem; parallel robots; linear actuators' orientations; assembly modes; general planar 3-RPR parallel mechanism; inertial measurement units; Cramér-Rao lower bound; static and dynamic experiments

---

## 1. Introduction

The direct kinematics problem is the problem of finding the actual position and orientation, also known as pose, of the moveable manipulator platform with respect to the fixed base platform from the active joints' coordinates. In general, this problem has multiple solutions. For example, for the general planar 3-RPR parallel mechanism, where three linear actuators, that is, active prismatic joints (P-joints), connect the passive revolute joints (R-joints) of the fixed base platform with those of the moveable manipulator platform, shown in Figure 1, up to six different poses of the manipulator platform are possible for a given set of linear actuators' lengths. These different poses that solve the direct kinematics problem are also known as assembly modes. However, the general questions that have to be answered for solving the direct kinematics problem in terms of control purposes are: (a) how many solutions exist for a given set of active joints' coordinates and (b) which one of them is the actual solution?

Many scientists have focused on answering these questions. The first question is basically solved by reducing the system of kinematic constraint equations to a univariate polynomial equation. Noncomplex solutions of this equation correspond to possible assembly modes of the parallel mechanism, that is, possible ways to assemble it. Among others, Gosselin et al. [1] introduced a polynomial formulation for the direct kinematics problem of the general planar 3-RPR parallel mechanism and concluded that for a given set of linear actuators' lengths, up to six real solutions can exist. The same result was independently achieved by Peisach, Pennock and Kassner and Wohlhart [2–4] and finally proved by Gosselin and Merlet [5]. Kong and Gosselin [6] even proposed a coordinate-free formulation to avoid dependencies on the chosen reference frame. In contrast to that, Collins [7] used Clifford algebra and Rojas et al. [8] introduced a method based on the bilateration problem to derive the polynomial formulation in a different manner. This distance-based method, however, can even yield two times more solutions compared to classical methods. For the special case where the three revolute base platform joints are aligned, only four solutions exist, see, for example, References [5,6,8].

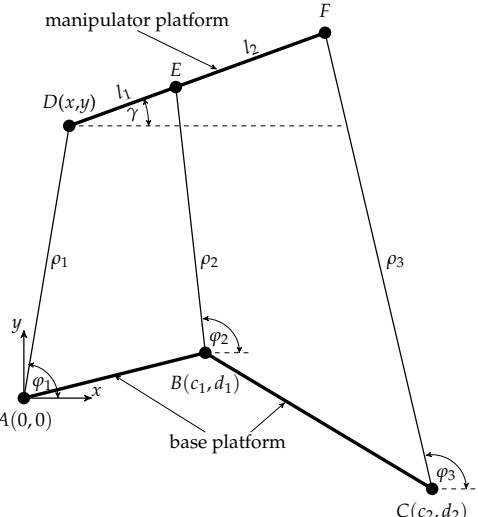

**Figure 1.** General planar 3-RPR parallel mechanism with the three base platform joints *A*, *B* and *C* and the three manipulator platform joints *D*, *E* and *F*. The pose of the manipulator platform is given by the position of joint *D* and the platform's orientation $\gamma$ with respect to the shown coordinate system.

Finding the univariate polynomial equation makes it possible to calculate all the possible solutions of the direct kinematics problem but it does not identify the actual pose of the manipulator platform. This can be done either by using additional numerical techniques such as Newton-Raphson algorithms with an initial pose estimation [9–15] to transform the system of nonlinear kinematic constraint equations into an explicit or linear problem where a closed-form solution can be found [16–23].

As the linear actuators' lengths are no generalized coordinates, they are only used because they are the active joints' coordinates. Due to the simple inverse kinematics of the general planar 3-RPR parallel mechanism with a unique solution, the linear actuators' lengths can be directly calculated when the manipulator platform's pose is known. This allows to use other coordinates that are more suitable for solving the direct kinematics problem and, afterwards, calculate the linear actuators' lengths from the obtained manipulator platform's pose. There are several advantages associated with avoiding the linear actuators lengths because (a) reference drives are required to derive the initial lengths, (b) absolute length sensors that do not need reference drives are very expensive and have a limited operation range, (c) possible deformations and backlashes in the linear actuators and joints cannot be determined, and, most importantly, (d) they do not provide a unique solution of the direct kinematics problem.

In order to avoid using the linear actuators' lengths for solving the direct kinematics, we proposed a new sensor concept where the manipulator platform's pose can be uniquely determined from the orientations provided by three inertial measurement units (IMUs) that were placed on top of the manipulator platform as well as on two of the linear actuators [24–26]. For measuring the manipulator platform's orientation, additional wiring effort is required that can cause workspace reductions due to the risk of link-wire interferences. In Reference [27], we therefore suggested using solely the three linear actuators' orientations for solving the direct kinematics problem and derived an analytical formulation that provides the two possible poses of the manipulator platform. Therewith, instead of having up to six assembly modes for the general planar 3-R$\underline{P}$R parallel mechanism when using the linear actuators' lengths that also cannot be found analytically (except for some special cases), we found an analytical expression to calculate them.

As the quality of the formulation's results mainly depends on the quality of the measured linear actuators' orientations, in this paper, we investigate the accuracy of our concept under static as well as dynamic conditions by performing several experiments on a new, specially designed prototype of a general planar 3-R$\underline{P}$R parallel mechanism. For measuring the linear actuators' orientations, we use inertial measurement units that provide linear accelerations and angular velocities of a rigid body in their three axes. Furthermore, we evaluate the maximum achievable accuracy of our formulation and investigate the effect of measurement errors on the calculated manipulator platform's pose by computing the Cramér-Rao lower bound and comparing the results with those of our experiments.

The remainder of this paper is as follows. In Section 2, classical methods for solving the direct kinematics problem of parallel mechanisms and especially the general planar 3-R$\underline{P}$R parallel mechanism are reviewed and their disadvantages are highlighted. In Section 3, we revisit the approach for calculating the number of possible assembly modes when solely the linear actuators' orientations are measured. In Section 4, we then derive the Cramér-Rao lower bound to estimate the variances of the calculated pose of the manipulator platform based on the variances of the linear actuators' orientations. In order to test our concept under static as well as dynamic conditions, in Section 5, we present the experimental results that were performed on a specially designed prototype of a general planar 3-R$\underline{P}$R parallel mechanism. Finally, a conclusion and evaluation is made in Section 6.

Throughout the paper, we use the following notation referring to Figure 1. The three base platform joints are denoted as *A*, *B* and *C* and the three manipulator platform joints as *D*, *E* and *F*. The body-fixed coordinate system of the base platform is located in joint *A* and the body-fixed coordinate system of the manipulator platform is located in joint *D*. The position of the manipulator platform with respect to the base platform is given by the coordinates *x* and *y* while the orientation of the manipulator platform is given by the angle $\gamma$. In the following, the manipulator platform's pose $\boldsymbol{p}$ with respect to the base platform is denoted by the position of the manipulator platform and its orientation:

$$\boldsymbol{p} = \begin{bmatrix} x & y & \gamma \end{bmatrix}^{\top}. \tag{1}$$

The coordinates of the two remaining base platform joints, *B* and *C*, are denoted as $c_1$ and $c_2$ in the *x*-axis and $d_1$ and $d_2$ in the *y*-axis. The three manipulator platform joints *D*, *E* and *F* are aligned and the distance between joint *D* and joint *E* is denoted as $l_1$, whereas the distance between joint *E* and joint *F* is denoted as $l_2$. The linear actuators' lengths are denoted as $\rho_1$, $\rho_2$ and $\rho_3$ and correspond to the distance between the joints *A* and *D*, *B* and *E*, as well as *C* and *D*, respectively. The linear actuators' orientation angles are denoted as $\varphi_1$, $\varphi_2$ and $\varphi_3$ and correspond to the angle between the *x*-axis and the first, second and third linear actuator, respectively.

## 2. Review of Classical Solutions for the Direct Kinematics Problem

In the literature, three methods are available for handling the direct kinematics problem of parallel mechanisms. Scientifically, the most interesting method is to derive the echelon form which contains all the solutions of the direct kinematics problem. Here, the system of kinematic constraint

equations is reduced to a univariate polynomial equation from which all the possible solutions are then derived. Noncomplex solutions of this equation correspond to possible assembly modes of the parallel mechanism, that is, modes for which the manipulator platform's pose satisfies the requirements of the active joints' coordinates as well as the closure conditions, or, in other words, possible solutions to assemble the parallel mechanism. The echelon form therewith allows to find all the possible solutions of the direct kinematics problem but it does not identify the actual or real pose of the manipulator platform. This problem can be solved by using one of the two following methods.

One possibility of finding the actual solution of the direct kinematics problem is to use iterative techniques such as Newton-Raphson procedures to solve the system of nonlinear kinematic constraint equations. These techniques require a good initial guess of the manipulator platform's pose on the one hand and a determinable pose that is sufficiently far away from a singular configuration on the other hand [28]. In this context, a singularity is a pose where the manipulator platform has at least one uncontrollable instantaneous degree of freedom leading to huge forces in the joints and the linear actuators, see, for example, References [29–31].

As an alternative to additional numerical procedures, in the third method, additional sensor information is used to transform the system of nonlinear kinematic constraint equations into an explicit or linear problem where a closed-form solution can be found. This method allows to find the actual pose of the manipulator platform uniquely and, compared to iterative methods, faster, more accurately and independently from initial pose estimations.

In the following, the three methods are reviewed and their complexity as well as remaining challenges are illustrated on the example of the general planar 3-R$\underline{P}$R parallel mechanism. As the planar equivalent to the Stewart-Gough platform, this mechanism has been investigated by several scientists in terms of direct kinematics [1–8,32], singularities [29–31,33–35] and control [36–38].

### 2.1. Analytical Solution

In this section, we review the classical method to derive the assembly modes of the general planar 3-R$\underline{P}$R parallel mechanism by following the method introduced by Gosselin et al. [1]. In contrast to the classical planar 3-R$\underline{P}$R parallel mechanism where the manipulator is illustrated as a triangle, we use the mechanism displayed in Figure 1. However, we show that by using the linear actuators' lengths, for this parallel mechanism, up to six solutions for the direct kinematics problem, that is, up to six assembly modes, exist.

The inverse kinematics of the general planar 3-R$\underline{P}$R parallel mechanism can be written as

$$\rho_1^2 = x^2 + y^2 \,, \tag{2}$$

$$\rho_2^2 = (x + l_1 \cos \gamma - c_1)^2 + (y + l_1 \sin \gamma - d_1)^2 \,, \tag{3}$$

$$\rho_3^2 = \left(x + (l_1 + l_2) \cos \gamma - c_2\right)^2 + \left(y + (l_1 + l_2) \sin \gamma - d_2\right)^2 \,. \tag{4}$$

By subtracting Equation (2) from Equation (3) and Equation (2) from Equation (4), we get

$$\rho_2^2 - \rho_1^2 = Rx + Sy + T \,, \tag{5}$$

$$\rho_3^2 - \rho_1^2 = Ux + Vy + W \tag{6}$$

with

$$
\begin{aligned}
R &= 2l_1 \cos \gamma - 2c_1 \,, \\
S &= 2l_1 \sin \gamma - 2d_1 \,, \\
T &= l_1^2 + c_1^2 + d_1^2 - 2l_1(c_1 \cos \gamma + d_1 \sin \gamma) \,, \\
U &= 2(l_1 + l_2) \cos \gamma - 2c_2 \,, \\
V &= 2(l_1 + l_2) \sin \gamma - 2d_2 \,, \\
W &= (l_1 + l_2)^2 + c_2^2 + d_2^2 - 2(l_1 + l_2)(c_2 \cos \gamma + d_2 \sin \gamma) \,.
\end{aligned}
\tag{7}
$$

From Equation (5), we get

$$
x = \frac{\rho_2^2 - \rho_1^2 - Sy - T}{R} \,,
\tag{8}
$$

and by inserting this result into Equation (6),

$$
y = \frac{R(\rho_3^2 - \rho_1^2 - W) - U(\rho_2^2 - \rho_1^2 - T)}{RV - SU} \,.
\tag{9}
$$

In the same manner, we get

$$
x = \frac{V(\rho_2^2 - \rho_1^2 - T) - S(\rho_3^2 - \rho_1^2 - W)}{RV - SU} \,.
\tag{10}
$$

In order to obtain a univariate equation in $\gamma$, inserting Equations (9) and (10) into Equation (2) gives us:

$$
\rho_1^2 = \frac{\left(V(\rho_2^2 - \rho_1^2 - T) - S(\rho_3^2 - \rho_1^2 - W)\right)^2}{(RV - SU)^2} + \frac{\left(R(\rho_3^2 - \rho_1^2 - W) - U(\rho_2^2 - \rho_1^2 - T)\right)^2}{(RV - SU)^2} \,.
\tag{11}
$$

By applying the Weierstrass substitution

$$
X = \tan \frac{\gamma}{2} \,, \quad \cos \gamma = \frac{1 - X^2}{1 + X^2} \,, \quad \sin \gamma = \frac{2X}{1 + X^2} \,,
\tag{12}
$$

We can get the sixth order polynomial in $X$, whose six possible solutions can be found numerically. In this context, Wenger et al. [33] investigated the situation where the term $RV - SU$ in Equations (9)–(11) becomes zero. Finally, we can substitute backwards and insert the solutions for $\gamma$ back into Equations (9) and (10) to obtain the position of the manipulator platform. However, there is no analytical solution for this problem available [8].

As an example, Figure 2 shows the six assembly modes of the general planar 3-R<u>P</u>R parallel mechanism when using the linear actuators' lengths. Here, we use the following parameters:

$$
c_1 = 40 \,\text{mm} \,, \quad d_1 = 10 \,\text{mm} \,, \quad l_1 = 25 \,\text{mm} \,, \quad c_2 = 90 \,\text{mm} \,, \quad d_2 = -20 \,\text{mm} \,, \quad l_2 = 35 \,\text{mm} \,.
\tag{13}
$$

With a given set of linear actuators' lengths

$$
\rho_1 = 80.6226 \,\text{mm} \,, \quad \rho_2 = 61.7931 \,\text{mm} \,, \quad \rho_3 = 82.9139 \,\text{mm} \,,
\tag{14}
$$

six assembly modes exist. Due to a faster calculation time, we derive the solution by using the method proposed by Rojas et al. [8]. The coordinates of point $D$, given by $x$ and $y$ and the orientation of the manipulator platform $\gamma$ for the six solutions are:

$$\begin{bmatrix} x_{\mathrm{I}} \\ y_{\mathrm{I}} \\ \gamma_{\mathrm{I}} \end{bmatrix} = \begin{bmatrix} 37.3098\,\mathrm{mm} \\ -71.4701\,\mathrm{mm} \\ -59.7539° \end{bmatrix}, \quad \begin{bmatrix} x_{\mathrm{II}} \\ y_{\mathrm{II}} \\ \gamma_{\mathrm{II}} \end{bmatrix} = \begin{bmatrix} -11.5040\,\mathrm{mm} \\ 79.7976\,\mathrm{mm} \\ -50.5183° \end{bmatrix}, \quad \begin{bmatrix} x_{\mathrm{III}} \\ y_{\mathrm{III}} \\ \gamma_{\mathrm{III}} \end{bmatrix} = \begin{bmatrix} 72.6382\,\mathrm{mm} \\ -34.9812\,\mathrm{mm} \\ 38.1265° \end{bmatrix}, \quad (15)$$

$$\begin{bmatrix} x_{\mathrm{IV}} \\ y_{\mathrm{IV}} \\ \gamma_{\mathrm{IV}} \end{bmatrix} = \begin{bmatrix} 10.0000\,\mathrm{mm} \\ 80.0000\,\mathrm{mm} \\ -20.0000° \end{bmatrix}, \quad \begin{bmatrix} x_{\mathrm{V}} \\ y_{\mathrm{V}} \\ \gamma_{\mathrm{V}} \end{bmatrix} = \begin{bmatrix} 36.0067\,\mathrm{mm} \\ 72.1354\,\mathrm{mm} \\ -9.0029° \end{bmatrix}, \quad \begin{bmatrix} x_{\mathrm{VI}} \\ y_{\mathrm{VI}} \\ \gamma_{\mathrm{VI}} \end{bmatrix} = \begin{bmatrix} 79.1195\,\mathrm{mm} \\ 15.4950\,\mathrm{mm} \\ 42.2360° \end{bmatrix}. \quad (16)$$

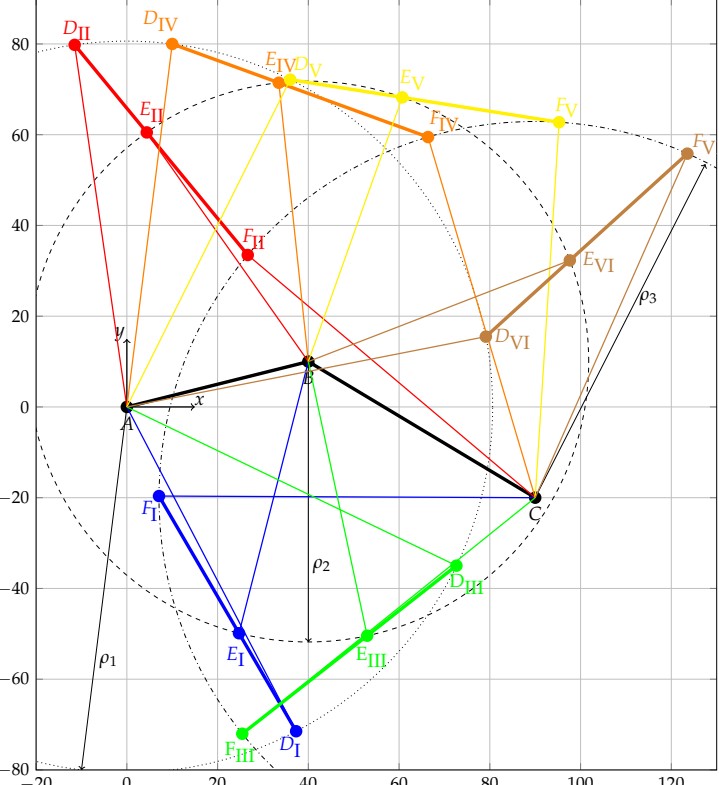

**Figure 2.** Assembly modes (shown in blue, red, green, orange, yellow and brown) for the manipulator platform of the general planar 3-R$\underline{\mathrm{P}}$R parallel mechanism when using the linear actuators' lengths $\rho_1$, $\rho_2$ and $\rho_3$ from Equation (14).

*2.2. Numerical Solution*

Since we are usually more interested in the actual manipulator platform's pose than in all of the possible poses, it is necessary to distinguish the actual pose from all the others. In the literature, there are numerous methods proposed that aim to find the actual pose of the parallel mechanism. Here, genetic algorithms [39–42], neuronal methods [43,44] and interval analysis methods [15,45] have to be mentioned. In fact, the most common numerical procedures for fast determination of the manipulator platform's pose are iterative techniques such as Newton-Raphson algorithms, see References [11,46–50]. Here, the inverse kinematic equations are used together with a pose estimate for iteratively solving these equations with a multi-dimensional Newton-Raphson algorithm.

All the iterative techniques have in common that they need a pose estimation. With the first guess, they calculate an error between the linear actuators' lengths that correspond to the pose estimate and the measured linear actuators' lengths. By using the measurement model, they vary the pose within several iterations to minimize this error. Different formulations and stop-criteria were proposed to

obtain the actual pose (see, for example, Reference [28]) but every iterative method depends on the quality of the first pose estimation. In fact, the pose the algorithm converges to is neither necessarily the actual pose due to the quality of the initial pose estimation nor the closest possible pose next to the initial estimate [28]. The iterative algorithm might also fail to converge in case of singularities [28,51]. In conclusion, the initial pose estimation influences both the pose the algorithm converges to and, not to be neglected, the computation time which corresponds to the number of iterations. For a static case, it is not possible to assure that the actual pose of the manipulator platform can be found at all because, depending on the initial pose estimate, all the solutions are possible. In fact, further information is still required to guarantee that the actual pose can be found.

Fortunately, for dynamic cases such as pose control, the initial guess can be improved during the sampling time and incorrect solutions can be removed. Since the converged pose is available for the previous set of linear actuators' lengths, this pose can be used as an initial guess together with the new set of lengths. Furthermore, the new solution has to be within some boundaries, based on the sampling time, maximum velocities and the latest pose [28].

As an example, we apply the Newton-Raphson algorithm to the general planar 3-R$\underline{\text{P}}$R parallel mechanism to compute the actual pose. In case a measurement model $h$ can be found that links the measurements $z$ with the manipulator platform's pose $p$, this pose can be found iteratively using the Newton-Raphson algorithm:

$$p(i) = p(i-1) + J_h\Big(z - h\big(p(i-1)\big)\Big) \tag{17}$$

where $i$ is the iteration step, $p(0)$ the initial pose estimate and $J_h$ the Jacobian of $h(p)$, which is

$$J_h\Big(h\big(p(i)\big)\Big) := J_h(i) = \frac{\mathrm{d}h(i)}{\mathrm{d}p(i)} \,. \tag{18}$$

The algorithm stops when the difference between the measurements $z$ and the results for the measurement model $h$ at the proposed pose $p(i)$ falls below a threshold value $\Lambda$, that is,

$$\left\| z - h\big(p(i)\big) \right\|_2 < \Lambda \,. \tag{19}$$

As the measurement model $h(p)$, the inverse kinematic Equations (2)–(4) are used.

The Newton-Raphson algorithm requires exact measurements and a good initial pose estimate. Again, the parameters from Equation (13) are used. As an example, the same linear actuators' lengths from Equation (14) are measured. Now, the Newton-Raphson algorithm is able to compute the pose that meets the conditions given by the measurement model. However, this can be any of the six possible solutions. It therewith depends on the quality of the initial pose estimate. For example, using

$$p(0) = \begin{bmatrix} 10.0000\,\text{mm} & 50.0000\,\text{mm} & 0.0000° \end{bmatrix}^\top \tag{20}$$

as initial pose estimate, after five iterations, the following solution is obtained:

$$\begin{bmatrix} x & y & \gamma \end{bmatrix}^\top = \begin{bmatrix} 10.0000\,\text{mm} & 80.0000\,\text{mm} & -20.0000° \end{bmatrix}^\top , \tag{21}$$

which corresponds to the fourth assembly mode and is shown in blue in Figure 3. But using

$$p(0) = \begin{bmatrix} 50.0000\,\text{mm} & 20.0000\,\text{mm} & 20.0000° \end{bmatrix}^\top , \tag{22}$$

after five iterations, leads to a different solution:

$$\begin{bmatrix} x & y & \gamma \end{bmatrix}^\top = \begin{bmatrix} 79.1195\,\text{mm} & 15.4950\,\text{mm} & 42.2360° \end{bmatrix}^\top, \tag{23}$$

shown in red in Figure 3. This solution corresponds to the fifth assembly mode. Furthermore, for the following initial pose estimate:

$$\boldsymbol{p}(0) = \begin{bmatrix} 0.0000\,\text{mm} & 0.0000\,\text{mm} & 0.0000° \end{bmatrix}^\top, \tag{24}$$

shown in green in Figure 3, the algorithm even fails to converge. Thus, it cannot be guaranteed that the actual pose can be found by using the Newton-Raphson algorithm because an appropriate initial pose estimate has to be provided. Otherwise, the algorithm can converge to the wrong solution or might even fail to converge.

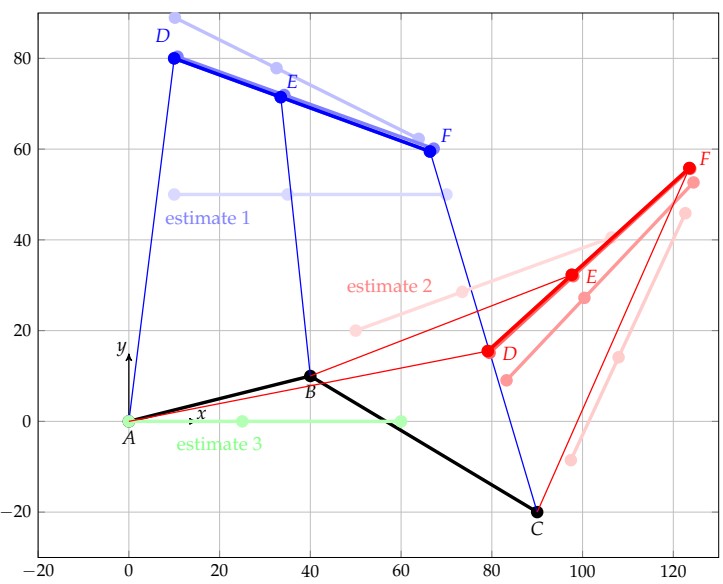

**Figure 3.** Solutions for the general planar 3-R$\underline{\text{P}}$R parallel mechanism when using a Newton-Raphson algorithm with the linear actuators' lengths: solution for $\begin{bmatrix} 10\,\text{mm} & 50\,\text{mm} & 0° \end{bmatrix}^\top$ (blue), for $\begin{bmatrix} 50\,\text{mm} & 20\,\text{mm} & 20° \end{bmatrix}^\top$ (red) and for $\begin{bmatrix} 0\,\text{mm} & 0\,\text{mm} & 0° \end{bmatrix}^\top$ (green) as initial pose estimates.

*2.3. Additional Sensor Solution*

As a matter of fact, analytical approaches where the inverse kinematic equations are used to obtain a univariate polynomial equation and iterative procedures are both vulnerable to measurement errors, calibration inaccuracies and sensor failure. If only the linear actuators' lengths are used, the manipulator platform's pose cannot be uniquely and unambiguously determined, neither for accurate measurements and optimal calibrated parallel mechanisms nor for perturbed measurements and calibrations. In order to overcome these disadvantages, it is possible to use sensor redundancy. By implementing further sensors, better and more reliable measurement results can be obtained and, in some cases, the actual pose can be determined without additional numerical procedures. In fact, the goal of redundant or auxiliary sensor concepts is to find an explicit or linear formulation for the manipulator platform's pose with the minimum number of sensor information.

The idea of using additional sensors to find the actual pose of the manipulator platform is based on the fact that the linear actuators' lengths are no minimal coordinates and, therewith, are not enough to find a unique solution for the direct kinematics problem. By implementing further sensors, it is possible to get more information about the system's state, reduce the complexity of the constraint equations and therewith, decrease the number of possible assembly modes until only one possible

pose of the manipulator platform remains. This allows to solve the direct kinematics equations in considerably less time, only limited by the sampling rate of the sensors but not the calculation time. The introduced information redundancy can later be used to increase the accuracy or to tolerate sensor failure or faulty sensor data, see, for example, Reference [17]. Furthermore, using additional sensors can even enable an auto-calibration of the parallel mechanism [28,52]. However, the type, number and location of the redundant sensors must be chosen very carefully to define a unique solution. Otherwise, it can cause additional problems such as workspace limitations due to the passive legs or joint arrangements, as mentioned in Reference [53]. Furthermore, different sensor types can even reduce the quality of the output by introducing time delays and unwarranted confidences. For example, trusting additional sensors with faulty measurements can lead to incorrect results or even prevent a result from being calculated.

Merlet [28] extensively discussed possible additional sensor concepts and Vertechy et al. [51] presented a very detailed, chronological review. Usually, length sensors and rotary sensors are used as additional sensors to derive the orientations of the linear actuators or passive legs in addition to the linear actuators' lengths, see, for example, References [16–23,51,53–68]. However, several other sensor types were proposed as additional sensors for solving the direct kinematics problem. For example, Baron et al. [69] suggested using a camera in addition to the linear actuators' lengths. For the 6-$\underline{R}$US Hexa-Robot, Hesselbach et al. [70] developed sensors that can be implemented in the passive joints. Inclination sensors can also be used. However, they are more often used for calibration purposes [71]. It can be noticed from the amount of papers dealing with the topic of additional sensor concepts for parallel mechanisms and especially the Stewart-Gough platform that the problem is quite complicated and the proposed solutions are not optimal. In fact, most of the concepts have one or more limitations. One drawback, for example, is the applicability, that is, some additional sensor concepts can only be used for parallel mechanisms with special architecture. The most common limitation is that the base and manipulator platform joints, respectively, should be coplanar, that is, lie on a plane, see, for example, References [18,20,57,58,65,68]. This architecture is often called nearly-general Stewart-Gough platform. Some other concepts require that two or more length sensors are connected to a common point or joint [18,19,63]. It furthermore stands out that all additional sensor concepts use at least one of the linear actuators' lengths and there are only few concepts where less than three linear actuators' lengths are used. In fact, none of the existing concepts completely renounce the lengths of the linear actuators and solely use other sensor information for solving the direct kinematics problem.

As an example for an additional sensor solution, we calculate the actual pose of the general planar 3-R$\underline{P}$R parallel mechanism by adding sensor information. One very common possibility to solve the direct kinematics problem is to add supplementary passive linear actuators that are equipped with length sensors. By coinciding the manipulator platform joints of one linear actuator with those of the supplementary linear actuator, see Figure 4, these joints' positions can be uniquely identified and the equations can be simplified to a closed-form solution. Here, for example, two passive linear actuators are added to the parallel mechanism. One is connected to the first and the other one to the third manipulator platform joint.

To the inverse kinematics of the general planar 3-R$\underline{P}$R parallel mechanism in Equations (2)–(4), two additional equations for the passive linear actuators can be added:

$$\rho_4^2 = (x - c_4)^2 + (y - d_4)^2\,, \tag{25}$$
$$\rho_5^2 = (x + (l_1 + l_2)\cos\gamma - c_5)^2 + (y + (l_1 + l_2)\sin\gamma - d_5)^2\,. \tag{26}$$

At first, the intersections of the two circles with the radii $\rho_1$ starting at point *A* and $\rho_4$ starting at point *G* shall be found, which, in this case, leads to two solutions. By subtracting Equation (2) from Equation (25), an equation for *x* can be obtained:

$$\rho_4^2 - \rho_1^2 = (x - c_4)^2 + (y - d_4)^2 - x^2 - y^2 = -2c_4 x - 2d_4 y + c_4^2 + d_4^2 \,, \tag{27}$$

$$\Longleftrightarrow x = \frac{\rho_1^2 - \rho_4^2 - 2d_4 y + c_4^2 + d_4^2}{2c_4} \,. \tag{28}$$

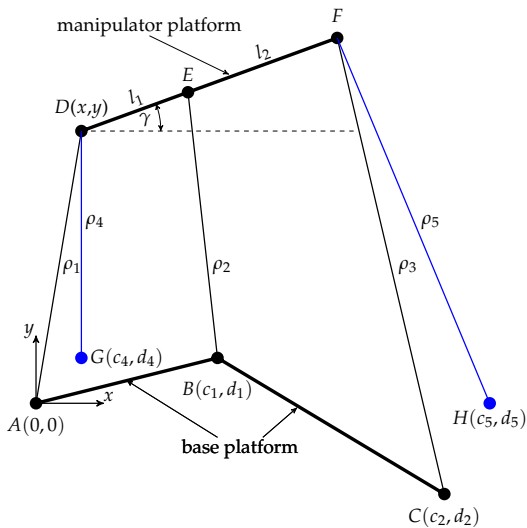

**Figure 4.** General planar 3-R$\underline{\text{P}}$R parallel mechanism with two additional passive linear actuators with the base platform joints $G$ and $H$. The active linear actuators are shown in black and the supplementary passive linear actuators are shown in blue.

With Equation (2), the two possible positions $x_{\text{I/II}}$ and $y_{\text{I/II}}$ of the manipulator platform are derived:

$$\rho_1^2 = \left( \frac{\rho_1^2 - \rho_4^2 - 2d_4 y + c_4^2 + d_4^2}{2c_4} \right)^2 + y^2 \tag{29}$$

$$\Longleftrightarrow y_{\text{I/II}} = \frac{\rho_1^2 d_4 - \rho_4^2 d_4 + c_4^2 d_4 + d_4^3}{2(c_4^2 + d_4^2)} \pm \frac{\sqrt{c_4^2(\rho_1^2 + 2\rho_1\rho_4 + \rho_4^2 - c_4^2 - d_4^2)(-\rho_1^2 + 2\rho_1\rho_4 - \rho_4^2 + c_4^2 + d_4^2)}}{2(c_4^2 + d_4^2)} \,, \tag{30}$$

$$x_{\text{I/II}} = \frac{\rho_1^2 - \rho_4^2 - 2d_4 y + c_4^2 + d_4^2}{2c_4} = \frac{c_4^4 + c_4^2 d_4^2 \mp d_4 \sqrt{c_4^2(c_4^2 + d_4^2)(4\rho_1^2 - c_4^2 - d_4^2)}}{2c_4(c_4^2 + d_4^2)} \,. \tag{31}$$

In a similar way, the two possible positions of point $F$ can be determined. Here, the following substitution is used:

$$\hat{x} = x + (l_1 + l_2) \cos \gamma \,, \qquad \hat{y} = y + (l_1 + l_2) \sin \gamma \,, \tag{32}$$

which reveals two solutions. The angle $\gamma$ can then be obtained by using the arc tangent with the known horizontal and vertical distances between the points $D$ and $F$.

As an example, Figure 5 shows the actual solution of the general planar 3-R$\underline{\text{P}}$R parallel mechanism when using the linear actuators' lengths and two additional lengths. Here, the parameters from Equation (13) are used together with

$$c_4 = 10 \,\text{mm} \,, \quad d_4 = 10 \,\text{mm} \,, \quad c_5 = 100 \,\text{mm} \,, \quad d_5 = 0 \,\text{mm} \,. \tag{33}$$

With the set of linear actuators' lengths from Equation (14) and

$$\rho_4 = 70.0000\,\text{mm}\,, \quad \rho_5 = 68.3222\,\text{mm}\,, \tag{34}$$

two solutions are obtained. The two possible coordinates of the manipulator platform, given by $x$ and $y$ and the orientation of the manipulator platform $\gamma$ are:

$$\begin{bmatrix} x_\text{I} \\ y_\text{I} \\ \gamma_\text{I} \end{bmatrix} = \begin{bmatrix} 10.0000\,\text{mm} \\ 80.0000\,\text{mm} \\ -20.0000° \end{bmatrix}, \quad \begin{bmatrix} x_\text{II} \\ y_\text{II} \\ \gamma_\text{II} \end{bmatrix} = \begin{bmatrix} 80.0000\,\text{mm} \\ 10.0000\,\text{mm} \\ -0.7883° \end{bmatrix}, \tag{35}$$

where the second solution is not possible because the distance between the points $D_\text{II}$ and $F_\text{II}$ does not satisfy the conditions given by $l_1$ and $l_2$. In fact,

$$\left\| D_\text{II} F_\text{II} \right\|_2 = \sqrt{(80.000\,\text{mm} - 167.7541\,\text{mm})^2 + (10.000\,\text{mm} - 8.7925\,\text{mm})^2}$$
$$= 87.7624\,\text{mm} \neq \sqrt{(l_1 + l_2)^2} = 60\,\text{mm}\,. \tag{36}$$

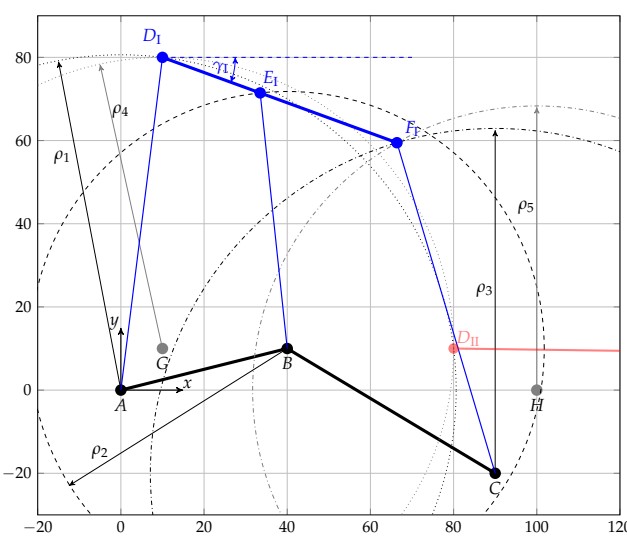

**Figure 5.** Actual solution (blue) for the general planar 3-R$\underline{\text{P}}$R parallel mechanism when using two additional lengths in addition to the linear actuators' lengths. The second solution is shown in red.

## 3. Assembly Modes when Using the Linear Actuators' Orientations

In the last section, we have seen that all the current concepts for solving the direct kinematics problem have several disadvantages as illustrated on the example of the general planar 3-R$\underline{\text{P}}$R parallel mechanism. In this section, we demonstrate that by using the three linear actuators' orientations, the solution of the direct kinematics problem of the general planar 3-R$\underline{\text{P}}$R parallel mechanism can be calculated analytically and a maximum of two instead of six assembly modes exist. Here, the elimination method described in Section 2.1 is used where the inverse kinematic equations are used to systematically eliminate unknown variables until a univariate equation is obtained.

For the general planar 3-R<u>P</u>R parallel mechanism shown in Figure 1, the inverse kinematics can be rewritten as

$$\underbrace{\tan \varphi_1}_{A} = \frac{y}{x} \,, \tag{37}$$

$$\underbrace{\tan \varphi_2}_{B} = \frac{y + l_1 \sin \gamma - d_1}{x + l_1 \cos \gamma - c_1} \,, \tag{38}$$

$$\underbrace{\tan \varphi_3}_{C} = \frac{y + (l_1 + l_2) \sin \gamma - d_2}{x + (l_1 + l_2) \cos \gamma - c_2} \,, \tag{39}$$

where we will be using the abbreviations $A$, $B$ and $C$ in the following. The angles $\varphi_1$, $\varphi_2$ and $\varphi_3$ are the three orientation angles of the linear actuators with respect to the base platform's $x$-axis and can be obtained, for example, from IMUs that are mounted on the linear actuators. Now, we can rewrite Equation (37):

$$y = Ax \,, \tag{40}$$

and use it in Equation (39):

$$C = \frac{Ax + (l_1 + l_2) \sin \gamma - d_2}{x + (l_1 + l_2) \cos \gamma - c_2} \,. \tag{41}$$

From this, we can derive an expression for $x$:

$$x = \frac{(l_1 + l_2)(- \sin \gamma + C \cos \gamma) - Cc_2 + d_2}{A - C} \,, \tag{42}$$

and from Equation (40), we can get an expression for $y$:

$$y = A \frac{(l_1 + l_2)(- \sin \gamma + C \cos \gamma) - Cc_2 + d_2}{A - C} \,. \tag{43}$$

In order to obtain a univariate equation in $\gamma$, we use Equations (42) and (43) with the remaining Equation (38) of the inverse kinematics:

$$B = \frac{y + l_1 \sin \gamma - d_1}{x + l_1 \cos \gamma - c_1} = \frac{A \frac{(l_1 + l_2)(- \sin \gamma + C \cos \gamma) - Cc_2 + d_2}{A - C} + l_1 \sin \gamma - d_1}{\frac{(l_1 + l_2)(- \sin \gamma + C \cos \gamma) - Cc_2 + d_2}{A - C} + l_1 \cos \gamma - c_1} \,. \tag{44}$$

Now, we have a univariate equation in $\gamma$ whose two possible solutions are given by

$$\gamma_{\mathrm{I}} = - \operatorname{atan} 2 \left( \frac{DH - E}{GI}, \frac{F + D}{G} \right) \,, \tag{45}$$

$$\gamma_{\mathrm{II}} = - \operatorname{atan} 2 \left( \frac{-DH - E}{GI}, \frac{F - D}{G} \right) \tag{46}$$

where

$$D = \sqrt{-I^2\left(-G + \frac{E^2}{I^4}\right)},\tag{47}$$

$$E = \Big(\big((-c_1 + c_2)C + c_1 A - d_2\big)B - (Ac_2 - d_1)C - A(d_1 - d_2)\Big)I^2,\tag{48}$$

$$
F = \Bigg(\bigg(\big(((-c_1 - c_2)l_1 - l_2 c_1)B + (d_1 - d_2)l_1 + (d_1 - d_2)l_2\big)C
$$

$$
+ c_1 l_1 B^2 - (d_1 - d_2)l_1 B + (c_2 l_1 + c_2 l_2)C^2\bigg)A^2 + \big((-c_1 + c_2)l_2 B^2 + d_1 l_2 B\big)C^2 - B^2 C d_2 l_2
$$

$$
+ \bigg(\big(((c_1 - c_2)l_1 + (c_1 - 2c_2)l_2)B - d_1 l_1 - d_1 l_2\big)C^2
$$

$$
+ \big(((-c_1 + c_2)l_1 + l_2 c_1)B^2 + ((d_1 + d_2)l_1 - (d_1 - 2d_2)l_2)B\big)C - B^2 d_2 l_1\bigg)A,\tag{49}
$$

$$G = (C^2 + 1)(A - B)^2 l_2^2 + l_1^2(B - C)^2(A^2 + 1) - 2l_1 l_2(B - C)(AC + 1)(A - B),\tag{50}$$

$$H = A(B - C)l_1 - C(A - B)l_2,\tag{51}$$

$$I = (A - B)l_2 - (B - C)l_1.\tag{52}$$

Finally, we can use the solutions for $\gamma$ in the Equations (42) and (43) to obtain the position of the manipulator platform. If we are also interested in the linear actuators' lengths, we can calculate them by using inverse kinematics, see Equations (2)–(4).

Figure 6a shows the two assembly modes of the general planar 3-R$\underline{\text{P}}$R parallel mechanism when the linear actuators' orientations are used. We use the same parameters as in Section 2. With a given set of linear actuators' orientation angles

$$\varphi_1 = 82.8750°, \qquad \varphi_2 = 96.0453°, \qquad \varphi_3 = 106.5502°,\tag{53}$$

the two assembly modes can be calculated. Using our method, we can find the following two solutions:

$$\begin{bmatrix} x_{\mathrm{I}} \\ y_{\mathrm{I}} \\ \gamma_{\mathrm{I}} \end{bmatrix} = \begin{bmatrix} 10.0000\,\text{mm} \\ 80.0000\,\text{mm} \\ -20.0000° \end{bmatrix}, \qquad \begin{bmatrix} x_{\mathrm{II}} \\ y_{\mathrm{II}} \\ \gamma_{\mathrm{II}} \end{bmatrix} = \begin{bmatrix} 24.2363\,\text{mm} \\ 193.8902\,\text{mm} \\ 104.5335° \end{bmatrix}.\tag{54}$$

As a second example, Figure 6b shows the two assembly modes of the general planar 3-R$\underline{\text{P}}$R parallel mechanism when the following linear actuators' orientation angles are used:

$$\varphi_1 = 82.8750°, \qquad \varphi_2 = 94.7360°, \qquad \varphi_3 = 101.0877°.\tag{55}$$

In this case, the following two solutions are obtained:

$$\begin{bmatrix} x_{\mathrm{I}} \\ y_{\mathrm{I}} \\ \gamma_{\mathrm{I}} \end{bmatrix} = \begin{bmatrix} 10.0000\,\text{mm} \\ 80.0000\,\text{mm} \\ 20.0000° \end{bmatrix}, \qquad \begin{bmatrix} x_{\mathrm{II}} \\ y_{\mathrm{II}} \\ \gamma_{\mathrm{II}} \end{bmatrix} = \begin{bmatrix} 10.2792\,\text{mm} \\ 82.2340\,\text{mm} \\ 23.6134° \end{bmatrix}.\tag{56}$$

Compared to the first example where the two solutions are far away from each other and the actual solution can be identified easily, in the second example, the two solutions are quite close to each other. This would make the differentiation of the actual solution more difficult, especially when the linear actuators' orientations are perturbed by measurement noise. In fact, when two linear actuators' orientations are identical, the root in Equation (47) becomes negative and no real solution exists. This also corresponds to a direct kinematics singularity.

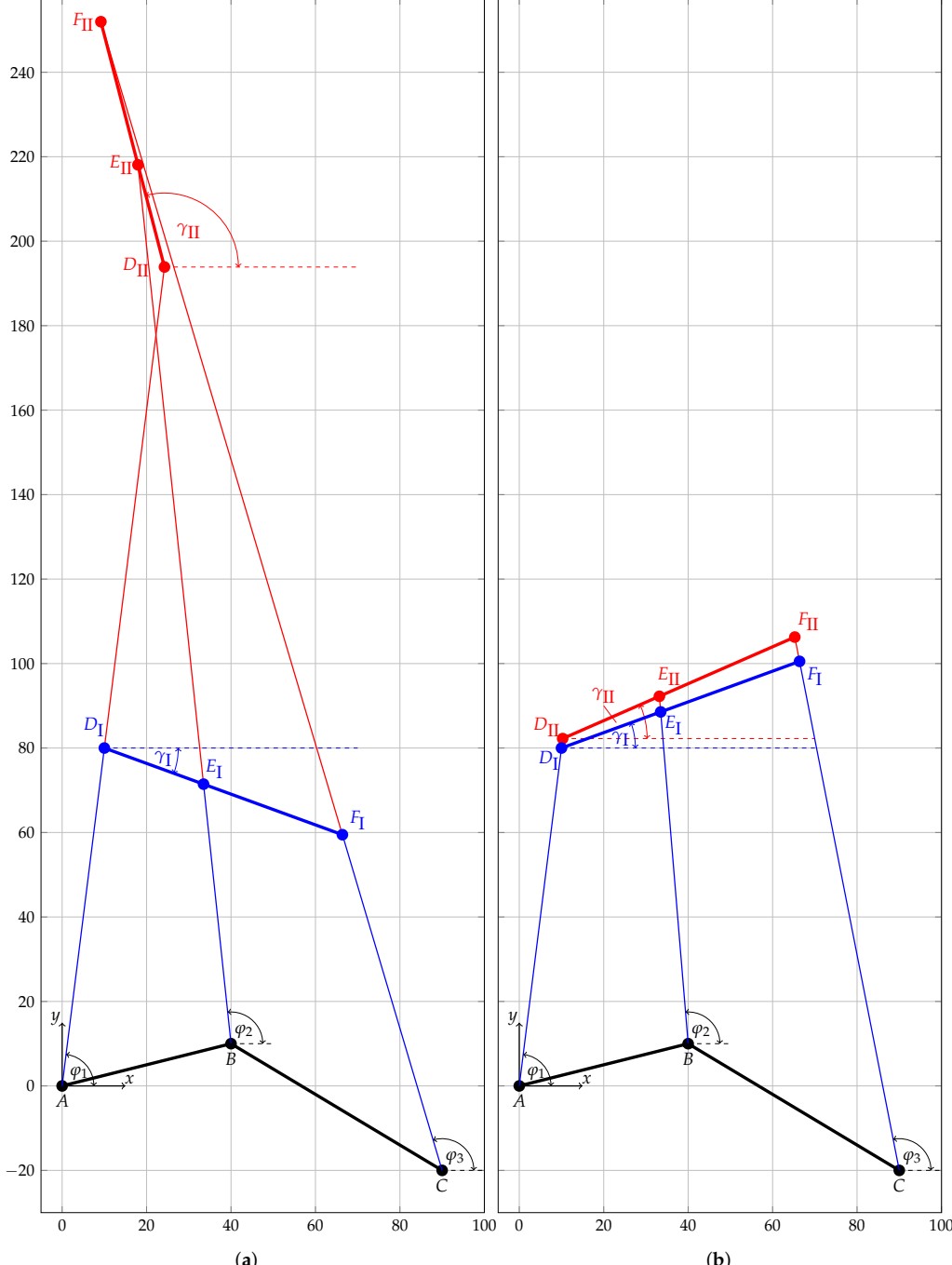

**Figure 6.** The two assembly modes (shown in blue and red) for the manipulator platform of the general planar 3-R$\underline{P}$R parallel mechanism when using the linear actuators' orientations: (**a**) results for $\varphi_1 = 82.8750°$, $\varphi_2 = 96.0453°$ and $\varphi_3 = 106.5502°$ and (**b**) results for $\varphi_1 = 82.8750°$, $\varphi_2 = 94.7360°$ and $\varphi_3 = 101.0877°$.

In general, it can be noticed that, in contrast to the usual six assembly modes, we only have two assembly modes when using the linear actuators' orientations. Furthermore, the assembly modes calculated from the linear actuators' orientations differ from those calculated from the liner actuators' lengths, compare Figures 2 and 6. Finally, the equations that were used for calculating the assembly modes are applicable to every type of general planar 3-R$\underline{P}$R parallel mechanisms and can be solved without any numerical methods. In contrast, when using the linear actuators' lengths,

there is, in general, no analytical equation available to calculate the assembly modes, see, for example, Reference [8].

## 4. Cramér-Rao Lower Bound

In order to evaluate the achievable accuracy of the presented approach, based on the expected variances of the linear actuators' orientations, the Cramér-Rao lower bound (CRLB) of the manipulator platform's pose can be computed and compared with the actually measured variances. The CRLB is an estimator that provides the lowest possible mean-squared error among all other estimators. Thus, it can be used to compare existing estimators or algorithms regarding their efficiency on the one hand and to estimate the impact of measurement errors on the calculated pose on the other hand.

By using the inverse kinematic Equations (37)–(39), we can find a relation between the measurement vector $z$, with

$$z = \begin{bmatrix} \varphi_1 & \varphi_2 & \varphi_3 \end{bmatrix}^\top , \tag{57}$$

and the pose $p$, with

$$p = \begin{bmatrix} x & y & \gamma \end{bmatrix}^\top , \tag{58}$$

that is given by the measurement model $h(p)$:

$$z = \begin{bmatrix} \operatorname{atan}\left(\frac{y}{x}\right) \\ \operatorname{atan}\left(\frac{y+l_1 \sin\gamma - d_1}{x+l_1 \cos\gamma - c_1}\right) \\ \operatorname{atan}\left(\frac{y+(l_1+l_2)\sin\gamma - d_2}{x+(l_1+l_2)\cos\gamma - c_2}\right) \end{bmatrix} =: h(p) . \tag{59}$$

Under the assumption that the measurement vector $z$ is zero-mean Gaussian distributed with its variances $\sigma^2(z_k)$, $k \in \{1, \ldots, 3\}$, that are stored in the covariance matrix $C$, with

$$C = \operatorname{diag}\left(\sigma^2(z_k)\right) , \tag{60}$$

we can calculate the CRLB as the inverse of the Fisher information matrix $F$. Its components can be determined as follows:

$$F_{k,l} = \frac{\partial z}{\partial p_k}^\top C^{-1} \frac{\partial z}{\partial p_l} + \frac{1}{2} \operatorname{tr}\left(C^{-1} \frac{\partial C}{\partial p_k} C \frac{\partial C}{\partial p_l}\right) , \tag{61}$$

with $k, l \in \{x, y, \gamma\}$, where $\frac{\partial z}{\partial p_k}$ are the components of the Jacobian $J_h$ of the measurement model $h(p)$:

$$J_h = \frac{\mathrm{d}h(p)}{\mathrm{d}p} = \begin{bmatrix} \frac{\partial h^\top(p)}{\partial x} & \frac{\partial h^\top(p)}{\partial y} & \frac{\partial h^\top(p)}{\partial \gamma} \end{bmatrix}^\top . \tag{62}$$

In general, the variances $\sigma^2(z_k)$ of the measurement vector $z$ are not constant and the trace in Equation (61) does not vanish so that we have to calculate the derivatives $\frac{\partial C}{\partial p_k}$. Assuming that the variances $\sigma^2(z_k)$, that is, $\sigma^2(\varphi_1)$, $\sigma^2(\varphi_2)$ and $\sigma^2(\varphi_3)$, only depend on the orientation angles $\varphi_1$, $\varphi_2$ and $\varphi_3$, the derivatives $\frac{\partial C}{\partial p_k}$ can be transformed as follows:

$$\frac{\partial C}{\partial p_k} = \frac{\partial C}{\partial z_k} \frac{\partial z_k}{\partial p_k} = \frac{\partial C}{\partial z_k} \frac{\partial h_k(p)}{\partial p_k} , \tag{63}$$

where $\frac{\partial h_k(p)}{\partial p_k}$ is a component of the Jacobian $J_h$ and $\frac{\partial C}{\partial z_k}$ is the derivative of the variance $\sigma^2(z_k)$ for the orientation angle $z_k$, that is, $\varphi_k$:

$$\frac{\partial C}{\partial z_k} = \frac{\partial \operatorname{diag}\left(\sigma^2(z_k)\right)}{\partial z_k} = \operatorname{diag}\left(\frac{\partial \sigma^2(z_k)}{\partial z_k}\right) , \tag{64}$$

and can be derived by experiments.

## 5. Experiments

### 5.1. Experimental Device

In order to investigate the accuracy of the direct kinematics solution under static as well as dynamic conditions, we perform experiments on a new, specially designed prototype, see Figure 7. It consists of three identical linear actuators that are connected on the one side to the base platform and on the other side to the manipulator platform. The base and the manipulator platform have integrated revolute joints and, furthermore, the possibility to vary the joints' positions. As linear actuators, we use Actuonix L16-100-35-P with a minimum length of 168 mm, a stroke length of 100 mm and an integrated potentiometer for measuring the current length. The linear actuators are equipped with IMUs to measure their orientation. Here, InvenSense MPU-9250 sensors are chosen as IMUs, where the accelerometer and the gyroscope values are used. An Arduino Mega board with an integrated data acquisition and pose calculation algorithm is mounted inside the experimental device. The Arduino Mega board is furthermore equipped with a display for showing the current pose and a motor shield for controlling the lengths of the linear actuators using a proportional-integral-derivative (PID) controller. For comparing the calculated manipulator platform's pose with the actual pose, we need an independent measurement system. Here, we use image processing to optically analyse the actual manipulator platform's pose, whose joints' positions are equipped with small red dots for optically tracking their position. The positions of the red dots' center points are therefore detected, stored and converted into the positions of the manipulator platform's joints from which the manipulator platform's pose can be calculated. For the experiments on the general planar 3-R$\underline{\text{P}}$R parallel mechanism, we use the following parameters for the base and manipulator platform's joints' coordinates according to Figure 1:

$$c_1 = 170 \,\text{mm}\,, \quad d_1 = 10 \,\text{mm}\,, \quad l_1 = 70 \,\text{mm}\,, \quad c_2 = 280 \,\text{mm}\,, \quad d_2 = -20 \,\text{mm}\,, \quad l_2 = 30 \,\text{mm}\,. \quad (65)$$

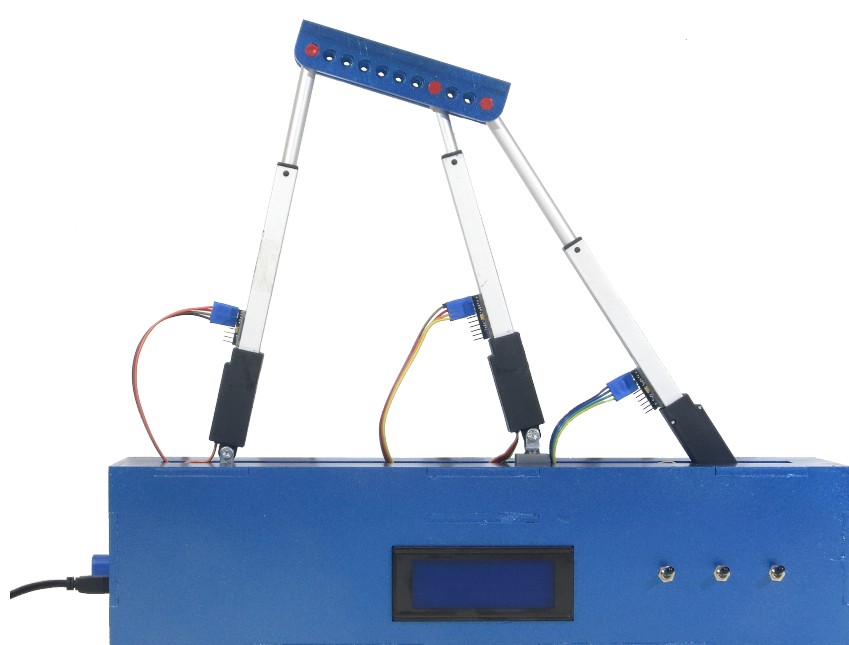

**Figure 7.** Experimental prototype of the general planar 3-R$\underline{\text{P}}$R parallel mechanism with inertial measurement units (IMUs) mounted on the linear actuators and an Arduino Mega with a display integrated in the base to calculate and show the two assembly modes of the manipulator platform.

## 5.2. Dynamic Orientation Measurement

The mechanism's $y$-axis corresponds to the negative gravity vector of the Earth $\boldsymbol{g}$. The IMUs are mounted on the linear actuators in the way that their $x$-axes are always parallel to the mechanism's $z$-axis. For static poses, it is therewith possible to obtain the orientation angles $\varphi_1$, $\varphi_2$ and $\varphi_3$ solely from the accelerometer values of the IMU, $a_{y,k}$ and $a_{z,k}$, where

$$\varphi_{k,\text{acc}} = \text{atan}\,2\left(a_{k,y}, a_{k,z}\right) . \tag{66}$$

However, when the manipulator platform moves and, therewith, the linear actuators move too, the accelerometer values do not provide accurate results, leading to faulty pose calculations. Robust methods for estimating the actual orientation angles of the linear actuators thus require the IMUs' gyroscope values $\omega_{k,x}$, $\omega_{k,y}$ and $\omega_{k,z}$ in addition to the accelerometer values. The orientation angle $\varphi_k$ of the $k$th linear actuator can be obtained, for example, by using a complementary filter with

$$\varphi_{k,\text{com}}(i) = \tau_{\text{t}}\left(\varphi_{k,\text{com}}(i-1) + \omega_{k,x}\Delta t\right) + (1 - \tau_{\text{t}})\,\varphi_{k,\text{acc}} , \tag{67}$$

where $i$ is the iteration step, $\tau_{\text{t}}$ is the ratio of the gyroscope and accelerometer values and $\Delta t$ is the time between two measurements. As initial orientation estimates, the results from the accelerometer values, that is, $\varphi_{k,\text{acc}}$, are used.

There are alternatives available for robustly and efficiently estimating the orientations based on IMU measurements including Kalman filtering, nonlinear complementary filters and quaternion based algorithms [72–75]. For the experiments, however, we choose the above introduced complementary filter with $\tau_{\text{t}} = 0.93$ for all the linear actuators. It shows fast responses to changes in the linear actuators' orientations as the gyroscope has a significantly higher impact than the accelerometer. In fact, especially for real-time applications on a low-memory computer, the complementary filter is recommendable because it shows similar accuracy with lower computational complexity compared to other filters. The time between two measurements $\Delta t$, which is the inverse of the sampling rate, mainly depends on the computational efficiency of the used algorithms, the programming and the processor. Throughout the experiments, we realized a sampling rate of 53.16 Hz that corresponds to a $\Delta t$ of 18.81 ms.

## 5.3. Accuracy of the Orientation Measurements

In order to investigate the dependency of the variances of the orientation angles $\sigma^2(\varphi_k)$ on the orientation angle $\varphi_k$ itself, we build a test bench, see Figure 8a, where we can mount different IMUs and vary the orientation angle in steps of $5°$. Here, we use an InvenSense MPU-9250 sensor which is rotated around its $z$-axis. For every orientation angle, we take 10,000 measurements with an Arduino Nano and calculate the orientation angle using the accelerometer and gyroscope values. Figure 8b shows the variances of the raw angle $\varphi_{\text{acc}}$, that is solely calculated from the accelerometer values and the filtered angle $\varphi_{\text{com}}$ for different orientation angles. For the raw orientation angle $\varphi_{\text{acc}}$, the variances lie between $0.0414°^2$ and $0.1609°^2$. In contrast to that, the filtered angle $\varphi_{\text{com}}$, shown in Figure 8c, has significantly smaller variances (27 to 38 times smaller) that lie between $0.0015°^2$ and $0.0042°^2$. In order to find a mathematical representation, we added a fifth-order polynomial fit with

$$\sigma^2(\varphi_{\text{acc}}) \approx a_0 + a_1\varphi_{\text{acc}} + a_2\varphi_{\text{acc}}^2 + a_3\varphi_{\text{acc}}^3 + a_4\varphi_{\text{acc}}^4 + a_5\varphi_{\text{acc}}^5 , \tag{68}$$

$$\sigma^2(\varphi_{\text{com}}) \approx a_0 + a_1\varphi_{\text{com}} + a_2\varphi_{\text{com}}^2 + a_3\varphi_{\text{com}}^3 + a_4\varphi_{\text{com}}^4 + a_5\varphi_{\text{com}}^5 , \tag{69}$$

where the constants $a_0$–$a_5$ are given in Table 1.

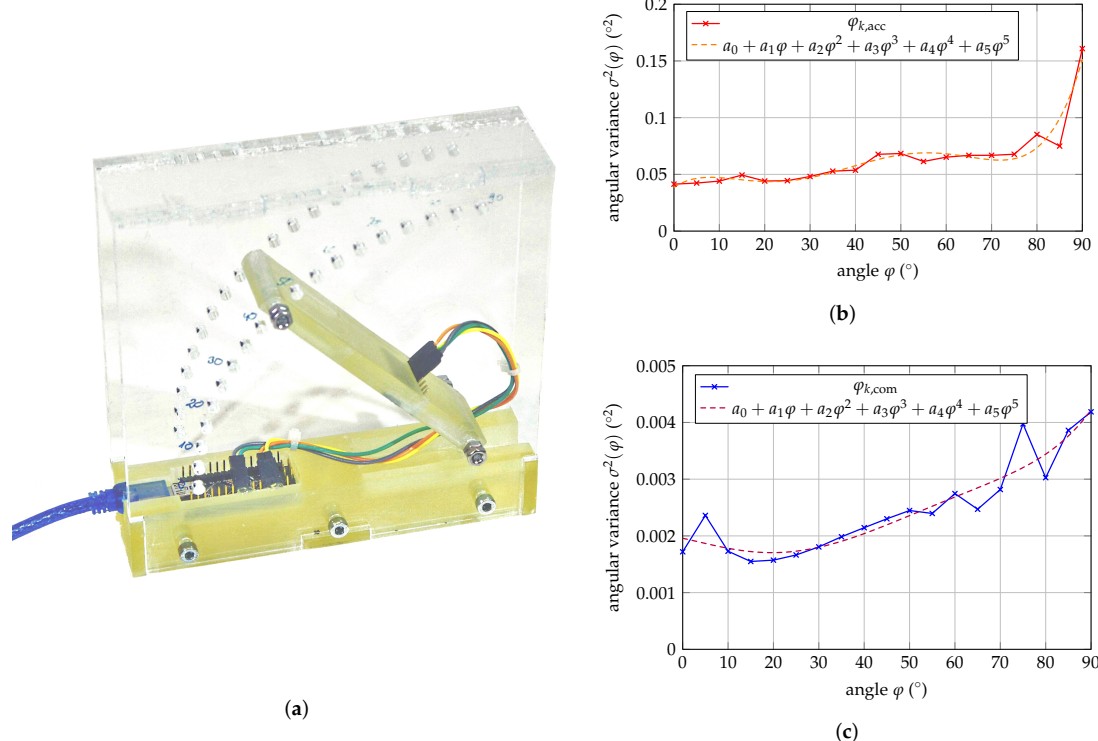

**Figure 8.** Experimental test bench to investigate IMUs on the dependency of the orientation angles' variances on the orientation angle (**a**). Experimental results of the InvenSense MPU-9250: (**b**) variances of the raw angle and suitable fifth-order polynomial fit and (**c**) variances of the filtered orientation angle and suitable fifth-order polynomial fit.

**Table 1.** Constants of the fifth-order polynomial for describing the orientation angle's variances of the raw and the filtered orientation angle as a function of the orientation angle itself.

| | $a_0$ | $a_1$ | $a_2$ | $a_3$ | $a_4$ | $a_5$ |
|---|---|---|---|---|---|---|
| $\varphi_{acc}$ | $3.8553 \times 10^{-2}$ | $2.7241 \times 10^{-1}$ | $-2.7631 \times 10^{-2}$ | $1.0431 \times 10^{-3}$ | $-1.5467 \times 10^{-5}$ | $7.8730 \times 10^{-8}$ |
| $\varphi_{com}$ | $1.9579 \times 10^{-3}$ | $-1.6773 \times 10^{-5}$ | $-5.1628 \times 10^{-7}$ | $5.0914 \times 10^{-8}$ | $-8.2501 \times 10^{-10}$ | $4.2277 \times 10^{-12}$ |

### 5.4. Accuracy of Static Pose Detections

As a first experiment on our general planar 3-R$\underline{P}$R parallel mechanism, we investigate how accurate the assembly modes can be obtained under static conditions when solely the linear actuators' orientations are used. We therefore investigate ten randomly chosen static poses of the manipulator platform where we take 500 measurements and calculate the two assembly modes from the measured linear actuators' orientation angles. In this context, we compare the accuracy for the assembly modes that can be obtained when raw orientation angles $\varphi_{acc}$ and filtered orientation angles $\varphi_{com}$ are used. In addition to this, we compare our experimental results with those provided by the CRLB. As the ground truth, we use the actual manipulator platform's pose whose joints' positions are optically analyzed by using image processing.

Table 2 shows the ten investigated, randomly chosen static poses. We choose the coordinates for the manipulator platform poses between 64.62 mm and 155.25 mm in the *x*-axis, between 157.14 mm and 216.06 mm in the *y*-axis and between $-20.45°$ and 15.84° for the platform orientation. Furthermore, Table 2 shows the mean values of the calculated poses after 500 measurements calculated from the raw orientation angles. First of all, it can be noticed that solution I, that is calculated from Equation (45), always corresponds to the actual pose while solution II, that is calculated from Equation (46), always

corresponds to the second assembly mode with higher $y$-coordinates. Second of all, it can be noticed that solution I and solution II are sufficiently far away from each other so that they can be distinguished unambiguously. Finally, when comparing the actual pose with the mean value of the calculated poses, it can be noticed that solution I has an offset error that varies from pose to pose between $-5.16\,\text{mm}$ and $0.11\,\text{mm}$ in the $x$-axis, between $-12.22\,\text{mm}$ and $1.03\,\text{mm}$ in the $y$-axis and between $-1.48°$ and $6.42°$ for the platform orientation. The offset errors do not seem to have any dependencies.

**Table 2.** Investigated static poses and mean values of the calculated poses (solution I and solution II) after 500 measurements obtained from the raw orientation angles. Dimensions are in mm and $°$.

| Pose | Actual Pose $\begin{bmatrix} x & y & \gamma \end{bmatrix}^\top$ | | | Solution I $\begin{bmatrix} x_\text{I} & y_\text{I} & \gamma_\text{I} \end{bmatrix}^\top$ | | | Solution II $\begin{bmatrix} x_\text{II} & y_\text{II} & \gamma_\text{II} \end{bmatrix}^\top$ | | |
|---|---|---|---|---|---|---|---|---|---|
| 1 | $\begin{bmatrix}146.76$ | $190.46$ | $14.01\end{bmatrix}^\top$ | $\begin{bmatrix}148.10$ | $190.89$ | $14.63\end{bmatrix}^\top$ | $\begin{bmatrix}309.13$ | $398.44$ | $-146.85\end{bmatrix}^\top$ |
| 2 | $\begin{bmatrix}90.71$ | $212.00$ | $-20.38\end{bmatrix}^\top$ | $\begin{bmatrix}94.96$ | $220.66$ | $-24.55\end{bmatrix}^\top$ | $\begin{bmatrix}146.43$ | $340.08$ | $-84.27\end{bmatrix}^\top$ |
| 3 | $\begin{bmatrix}137.55$ | $206.21$ | $-7.71\end{bmatrix}^\top$ | $\begin{bmatrix}137.68$ | $206.42$ | $-6.67\end{bmatrix}^\top$ | $\begin{bmatrix}250.39$ | $375.34$ | $-107.89\end{bmatrix}^\top$ |
| 4 | $\begin{bmatrix}155.25$ | $191.61$ | $15.72\end{bmatrix}^\top$ | $\begin{bmatrix}155.41$ | $190.95$ | $17.04\end{bmatrix}^\top$ | $\begin{bmatrix}322.82$ | $396.65$ | $-151.26\end{bmatrix}^\top$ |
| 5 | $\begin{bmatrix}123.65$ | $211.69$ | $-11.96\end{bmatrix}^\top$ | $\begin{bmatrix}124.50$ | $211.93$ | $-11.72\end{bmatrix}^\top$ | $\begin{bmatrix}217.26$ | $369.73$ | $-99.65\end{bmatrix}^\top$ |
| 6 | $\begin{bmatrix}69.22$ | $215.68$ | $-12.16\end{bmatrix}^\top$ | $\begin{bmatrix}74.53$ | $228.82$ | $-18.83\end{bmatrix}^\top$ | $\begin{bmatrix}124.70$ | $382.65$ | $-90.18\end{bmatrix}^\top$ |
| 7 | $\begin{bmatrix}107.01$ | $190.51$ | $0.71\end{bmatrix}^\top$ | $\begin{bmatrix}107.98$ | $192.76$ | $-0.73\end{bmatrix}^\top$ | $\begin{bmatrix}219.71$ | $392.17$ | $-120.73\end{bmatrix}^\top$ |
| 8 | $\begin{bmatrix}64.62$ | $186.05$ | $15.84\end{bmatrix}^\top$ | $\begin{bmatrix}66.82$ | $191.85$ | $10.75\end{bmatrix}^\top$ | $\begin{bmatrix}156.08$ | $448.07$ | $-144.02\end{bmatrix}^\top$ |
| 9 | $\begin{bmatrix}125.21$ | $161.73$ | $13.32\end{bmatrix}^\top$ | $\begin{bmatrix}125.31$ | $162.46$ | $13.50\end{bmatrix}^\top$ | $\begin{bmatrix}276.07$ | $357.91$ | $-153.10\end{bmatrix}^\top$ |
| 10 | $\begin{bmatrix}132.37$ | $157.14$ | $8.30\end{bmatrix}^\top$ | $\begin{bmatrix}132.71$ | $158.64$ | $7.45\end{bmatrix}^\top$ | $\begin{bmatrix}284.36$ | $339.89$ | $-142.95\end{bmatrix}^\top$ |

Figure 9 shows the position and orientation errors between the investigated static poses and solution I that was obtained experimentally from the IMUs' values with 500 repetitions as boxplots. The results from the raw accelerometer values are shown in red and the results from the complementary filtered orientation angles are shown in blue. Comparing the results from the raw accelerometer values with those obtained from the filtered orientation angles, it can be noticed that both show similar offset errors but, most importantly, the results for the filtered orientation angles have significantly lower variances. In fact, throughout the ten investigated poses, the variances of the position and orientation errors obtained with the raw accelerometer values are approximately 27 times higher than those obtained with the filtered orientation angles. This applies for all axes (8.3 to 56.1 times higher for the $x$-axis, 14.2 to 37.5 times higher for the $y$-axis and 14.2 to 33.3 times higher for the platform orientation). From the results shown in Figure 9, it can also be noticed that the variances in the axes are not constant and show dependencies on the position and orientation of the manipulator platform. In fact, the best results for the raw accelerometer values were obtained for the poses 9 and 10 where the position and orientation errors show variances of only $0.19\,\text{mm}^2$ to $0.34\,\text{mm}^2$ in the $x$-axis, $1.60\,\text{mm}^2$ to $1.63\,\text{mm}^2$ in the $y$-axis and $2.43°^2$ to $2.96°^2$ for the platform orientation. In contrast to that, poses 2 and 6 show the highest variances for the raw accelerometer values with $5.07\,\text{mm}^2$ to $7.33\,\text{mm}^2$ in the $x$-axis, $65.15\,\text{mm}^2$ to $74.55\,\text{mm}^2$ in the $y$-axis and $20.36°^2$ to $22.71°^2$ for the platform orientation. The same applies for the filtered orientation angles.

In conclusion, the manipulator platform's pose can be obtained quite accurately with only small offset errors. Nevertheless, the variances obtained for the raw accelerometer values are very high but can be significantly improved when using the filtered orientation angles instead. Here, variances between $0.006\,\text{mm}^2$ and $0.155\,\text{mm}^2$ for the $x$-axis, between $0.051\,\text{mm}^2$ and $2.450\,\text{mm}^2$ for the $y$-axis and between $0.073°^2$ and $0.743°^2$ can be obtained. In comparison to the other poses, the variances for poses 2 and 6 are comparably higher. One possible reason may be that only poses 2, 6 and 8, the orientation angle $\varphi_2$ is above $90°$, whereas $\varphi_2$ is below $90°$ for the other poses. However, pose 8 does not show the high variances that would be expected.

The variances of the position and orientation errors depend on the variances of the measurements and, more importantly, the amplification of the solution formulation. The CRLB allows to calculate the lower bound of the variances that we can expect for a specific pose based on the variances of the measurements, that is, the orientation angles. Therewith, we can compare our experimental results for each pose with those calculated by the CRLB. Figure 10 shows the position and orientation errors obtained experimentally from the filtered orientation angles in blue and the simulated position and orientation errors calculated with the CRLB in purple. The CRLB only requires the actual pose, the measurement model and the variances of the measurements. Here, we use the polynomial in Equation (69) to estimate the variances of the filtered orientation angles.

Table 3 shows the variances and CRLB's results for the first five static poses when using raw orientation angles and when using filtered orientation angles. From the first five investigated static poses, it can already be noticed that the variances of the position and orientation errors obtained from experiments correspond with those calculated by the CRLB. The same applies for the poses six to ten (not displayed). For all the poses, the difference between the experimental results from the filtered orientation angles and the CRLB's results are very small and do not exceed $0.15\,\text{mm}^2$ in the $x$-axis, $0.28\,\text{mm}^2$ in the $y$-axis and $0.32^{\circ 2}$ for the platform orientation. When using the CRLB together with the polynomial in Equation (68) as the variances of the measurements, we can estimate the variances of the position and orientation errors of the results obtained with the raw accelerometer values similarly accurate (not displayed in Figure 10). Here, the difference between the raw experimental results and the CRLB's results are not higher than $1.42\,\text{mm}^2$ in the $x$-axis, $8.00\,\text{mm}^2$ in the $y$-axis and $-5.82^{\circ 2}$ for the platform orientation.

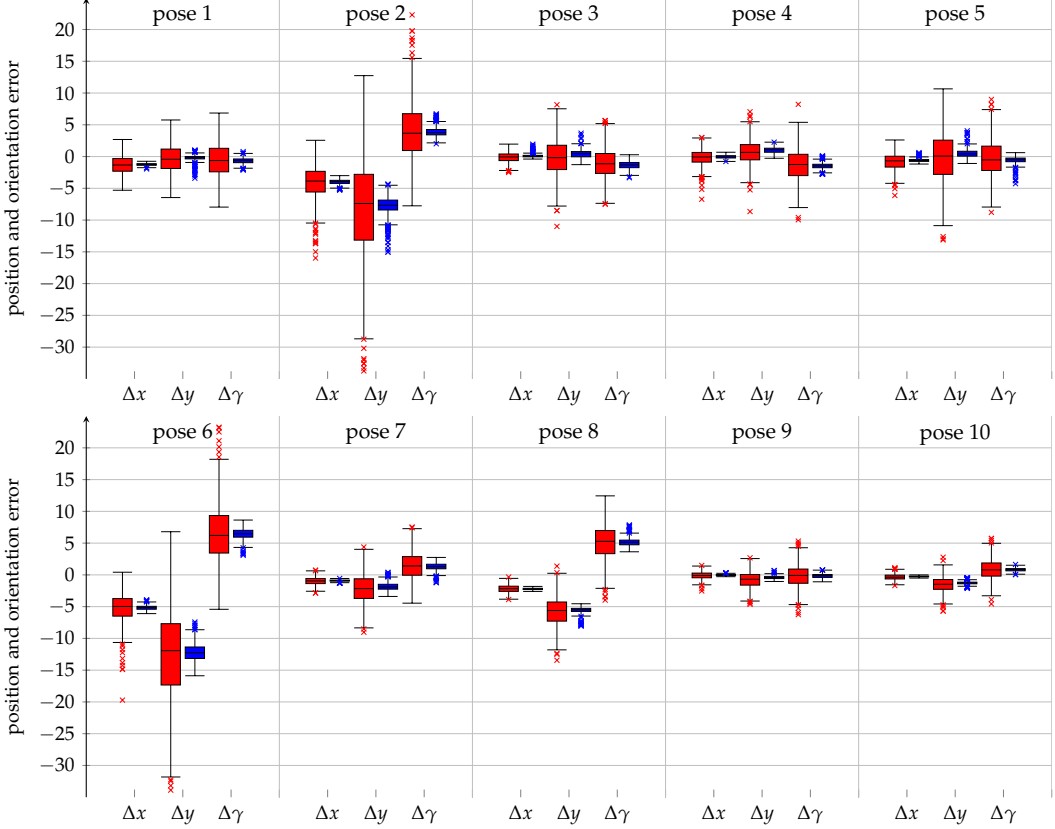

**Figure 9.** Results for the ten investigated static poses with 500 repetitions obtained experimentally from the raw accelerometer values (red) and the filtered orientation angles (blue). The errors in each axis, $\Delta x$, $\Delta y$ and $\Delta \gamma$, are displayed in a boxplot. Dimensions are in mm and $^\circ$. The box corresponds to the area in which the middle 50% of the errors lie while the whiskers indicate the area in which the middle 99.3% of the errors lie.

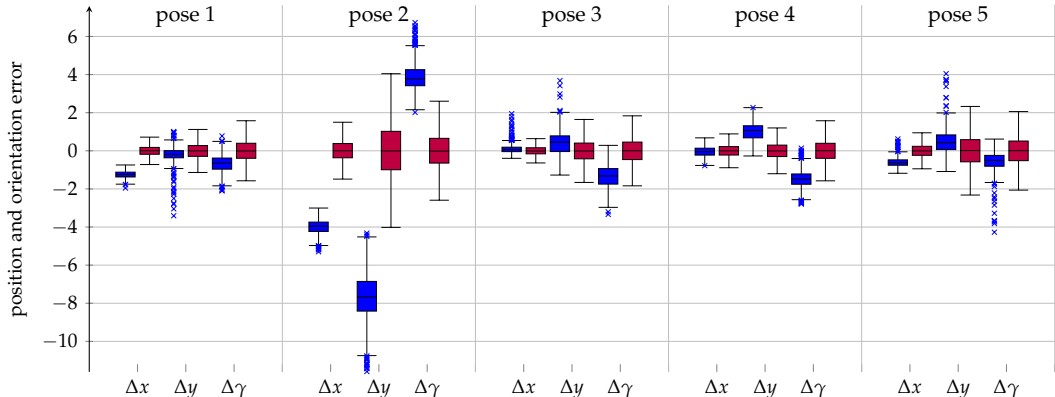

**Figure 10.** Results for the first five investigated static poses with 500 repetitions obtained experimentally from the filtered orientation angles (blue) and by simulation using the corresponding Cramér-Rao lower bound (CRLB) (purple). The errors in each axis, $\Delta x$, $\Delta y$ and $\Delta \gamma$, are displayed in a boxplot. Dimensions are in mm and °. The box corresponds to the area in which the middle 50% of the errors lie while the whiskers indicate the area in which the middle 99.3% of the errors lie.

**Table 3.** Variances and results for the Cramér-Rao lower bound for the first five static poses when using raw orientation angles and when using filtered orientation angles. The variances are displayed as $\begin{bmatrix} \sigma^2(x) & \sigma^2(y) & \sigma^2(\gamma) \end{bmatrix}^\top$. Dimensions are in mm$^2$ and °$^2$.

| Pose | Variances for Raw Orientation Angles | | Variances for Filtered Orientation Angles | |
|---|---|---|---|---|
| | Experiments | CRLB | Experiments | CRLB |
| 1 | $\begin{bmatrix} 1.8895 \\ 4.4329 \\ 6.9274 \end{bmatrix}$ | $\begin{bmatrix} 1.6150 \\ 4.1013 \\ 7.6160 \end{bmatrix}$ | $\begin{bmatrix} 0.0337 \\ 0.2244 \\ 0.2335 \end{bmatrix}$ | $\begin{bmatrix} 0.0715 \\ 0.1697 \\ 0.3470 \end{bmatrix}$ |
| 2 | $\begin{bmatrix} 7.3331 \\ 65.1472 \\ 20.3617 \end{bmatrix}$ | $\begin{bmatrix} 7.7918 \\ 57.1507 \\ 23.6220 \end{bmatrix}$ | $\begin{bmatrix} 0.1493 \\ 2.4503 \\ 0.6109 \end{bmatrix}$ | $\begin{bmatrix} 0.2968 \\ 2.2853 \\ 0.9335 \end{bmatrix}$ |
| 3 | $\begin{bmatrix} 0.6376 \\ 7.9631 \\ 6.0536 \end{bmatrix}$ | $\begin{bmatrix} 1.2437 \\ 9.4596 \\ 9.7948 \end{bmatrix}$ | $\begin{bmatrix} 0.0767 \\ 0.4116 \\ 0.3189 \end{bmatrix}$ | $\begin{bmatrix} 0.0582 \\ 0.3787 \\ 0.4445 \end{bmatrix}$ |
| 4 | $\begin{bmatrix} 1.5456 \\ 3.5633 \\ 5.6761 \end{bmatrix}$ | $\begin{bmatrix} 2.5668 \\ 4.9758 \\ 7.7996 \end{bmatrix}$ | $\begin{bmatrix} 0.0594 \\ 0.1653 \\ 0.2352 \end{bmatrix}$ | $\begin{bmatrix} 0.1070 \\ 0.2006 \\ 0.3483 \end{bmatrix}$ |
| 5 | $\begin{bmatrix} 1.6415 \\ 14.4296 \\ 8.1143 \end{bmatrix}$ | $\begin{bmatrix} 3.0573 \\ 18.2156 \\ 13.9312 \end{bmatrix}$ | $\begin{bmatrix} 0.0593 \\ 0.4560 \\ 0.3463 \end{bmatrix}$ | $\begin{bmatrix} 0.1263 \\ 0.7334 \\ 0.5923 \end{bmatrix}$ |

In conclusion, by only knowing the measurement variances of the IMUs, it is possible to predict the manipulator platform's variances very accurately for any pose in the workspace without experiments at all. As the experimental results match the CRLB's results, we can furthermore conclude that the solution formulation proposed in Section 4 is the optimal estimator with the lowest amplification of measurement variances on the position and orientation variances. In the measurement model for the CRLB, we assumed the measurement error to be zero-mean Gaussian. The experiments indicate that this is not true. By also including these offset errors and the nonlinearity of the IMUs into the measurement model, it would be possible to predict the offset error of the manipulator platform's pose in addition to its variances. As an alternative, in order to realize zero-mean Gaussian measurement errors, it is also be possible to eliminate the offset errors and the nonlinearity of the IMUs by doing further calibrations.

*5.5. Comparing Analytic Orientation-Based Results with Iterative Length-Based Results for Static Pose Detections*

In Section 2, we reviewed classical methods for solving the direct kinematics problem of the general planar 3-RPR parallel mechanism and mentioned that iterative methods like the Newton-Raphson algorithm are most often used for finding the actual pose of the manipulator platform from the linear actuators' lengths. The accuracy mainly depends on the initial estimate and the accuracy of the measured linear actuators' lengths. The linear actuators that are used in our prototype of the general planar 3-RPR parallel mechanism have integrated potentiometers. Consequently, the lengths are measured indirectly with the problem that the actual lengths of the linear actuators are not measured and rely on the linearities of the potentiometers. In addition, the analog inputs of the Arduino Mega board have a limited resolution of 10 bit leading to a maximum resolution of 0.0977 mm/bit for the linear actuators' lengths (stroke length of the linear actuators divided by the resolution of Arduino Mega). Nevertheless, the obtained linear actuators' lengths can be used together with an initial estimate and the Newton-Raphson algorithm to calculate the actual pose of the manipulator platform. In order to evaluate the quality of the measured linear actuators' lengths and to guarantee convergence of the Newton-Raphson algorithm, we used the actual pose as initial estimate. Table 4 shows the mean offset errors of the Newton-Raphson algorithm for the measured linear actuators' lengths. Furthermore, it shows the mean offset errors for the ten static poses obtained with the analytic formulation proposed in Section 3 and the raw orientation angles.

**Table 4.** Investigated static poses and mean offset errors $\Delta x$, $\Delta y$ and $\Delta \gamma$ of the analytic, orientation-based formulation and the iterative length-based solution (Newton-Raphson algorithm). Dimensions are in mm and $^\circ$. Poses, where the algorithm fails to converge are indicated by a $--$.

| Pose | Actual Pose $\begin{bmatrix} x & y & \gamma \end{bmatrix}^\top$ | | | Offset Error Solution I $\begin{bmatrix} \Delta x & \Delta y & \Delta \gamma \end{bmatrix}^\top$ | | | Offset Error Newton-Raphson Algorithm $\begin{bmatrix} \Delta x & \Delta y & \Delta \gamma \end{bmatrix}^\top$ | | |
|---|---|---|---|---|---|---|---|---|---|
| 1 | 146.76 | 190.46 | 14.01 | −1.25 | −0.23 | −0.66 | 8.18 | −9.22 | 4.09 |
| 2 | 90.71 | 212.00 | −20.38 | −3.99 | −7.83 | 3.88 | −1.85 | −2.77 | 1.58 |
| 3 | 137.55 | 206.21 | −7.71 | 0.11 | 0.39 | −1.35 | −0.42 | −3.79 | 1.38 |
| 4 | 155.25 | 191.61 | 15.72 | −0.05 | 1.03 | −1.48 | 2.77 | −4.41 | 0.78 |
| 5 | 123.65 | 211.69 | −11.96 | −0.59 | 0.51 | −0.60 | −0.82 | −2.92 | 1.36 |
| 6 | 69.22 | 215.68 | −12.16 | −5.16 | −12.22 | 6.42 | 1.32 | −3.69 | 2.25 |
| 7 | 107.01 | 190.51 | 0.71 | −0.90 | −1.85 | 1.26 | −3.12 | 0.21 | −1.08 |
| 8 | 64.62 | 186.05 | 15.84 | −2.21 | −5.55 | 5.16 | −32.48 | 5.86 | −11.14 |
| 9 | 125.21 | 161.73 | 13.32 | −0.02 | −0.41 | −0.15 | −− | −− | −− |
| 10 | 132.37 | 157.14 | 8.30 | −0.25 | −1.26 | 0.82 | −− | −− | −− |

From Table 4, it can be observed that, except for the poses 2 and 6, the mean error of the Newton-Raphson algorithm is significantly higher than the mean error of the analytic formulation. In fact, the mean errors of the Newton-Raphson algorithm are 1.5 to even 20 times higher than the mean errors of the analytic formulation. The mean errors spread between −32.48 mm and 8.18 mm in the *x*-axis, between −9.22 mm and 5.86 mm in the *y*-axis and between −11.14° and 4.09° for the platform orientation. For the poses 9 and 10, the Newton-Raphson algorithm even converged to a completely wrong solution although the actual pose is used as the initial pose estimate. In contrast, for poses 2 and 6, the Newton-Raphson algorithm shows more accurate results than the analytic formulation. The mean errors in the calculated poses indicate that there is an offset between the actual and the measured linear actuators' lengths that needs to be removed by calibration. By comparing the actual lengths with the measured lengths, however, only small errors in the lengths measurements

were recognized ($\pm 1.5$ mm). Other than for the linear actuators' orientations and consequently the results for the analytic formulation, the variances for the Newton-Raphson algorithm are nearly zero (0.21 mm$^2$ in the $x$-axis, 0.09 mm$^2$ in the $y$-axis and 0.06$^{\circ 2}$ for the platform orientation) since the lengths do not change under static conditions and the potentiometers' readings only differ by $\pm 1$ bit. This indicates that if the lengths are measured correctly and with a sufficiently high resolution, the manipulator platforms' pose can be found with the Newton-Raphson algorithm more robustly than from the unfiltered orientation angles. In the current form, that is, using the linear actuators' potentiometer values, only slightly lower variances as for the filtered orientation angles can be obtained. However, the Newton-Raphson algorithm requires at least three to five iterations to converge, whereas the analytic formulation provides an explicit formulation without any iteration steps. Furthermore, if the initial pose estimate is changed away from the actual pose, the required number of iterations increase and we cannot guarantee that the Newton-Raphson algorithm will always converge to the correct solution.

In conclusion, the Newton-Raphson algorithm together with the linear actuators' lengths shows higher offset errors but lower variances than the analytic formulation where solely the linear actuators' orientations are used. However, for pose detections where no accurate initial pose estimate can be provided, for example, in the beginning of an experiment or after restarting the system, the convergence of the Newton-Raphson algorithm cannot be guaranteed.

### 5.6. Accuracy of Dynamic Pose Detections

As a second experiment on our general planar 3-R<u>P</u>R parallel mechanism, we investigate how accurate the manipulator platform's pose can be obtained under dynamic conditions when solely the linear actuators' orientations are used. Therefore, we continuously move the manipulator platform dynamically by changing the linear actuators' lengths adequately using a PID controller that minimizes the differences between the target and the actual lengths. We let the linear actuators run with 12 V leading to higher velocities ($\pm 40$ mm/s and $\pm 15^\circ$/s). During the experiment, we measure and filter the linear actuators' orientations with the maximum possible sampling rate, that still is 53.16 Hz and calculate the two assembly modes using the formulation proposed in Section 3. As the ground truth, we again use image processing to optically analyse the actual manipulator platform's pose, whose joints' positions are equipped with small red dots for optically tracking their position (the images are recorded with 30 fps). Figure 11 shows the trajectories of the manipulator platform's joints in blue, red and green, respectively, during the dynamic experiment. The entire dynamic experiment is also shown in the video of the Supplementary Material.

Figure 12 shows the manipulator platform's pose during the dynamic experiment calculated from the raw (red) and the filtered (blue) linear actuators' orientations. As reference (black), the positions and orientations calculated from the optically analysed manipulator platform joints are displayed. During the experiment, the manipulator platform's pose ranges between 97.6 mm and 154.5 mm in the $x$-axis, between 177.1 mm and 219.1 mm in the $y$-axis and between 11.6$^\circ$ and 24.7$^\circ$ for the platform orientation. Here, we only use solution I of the proposed formulation calculated from Equation (45). Solution II range between 165.9 mm and 333.7 mm in the $x$-axis, between 332.2 mm and 492.6 mm in the $y$-axis and between $-179.9^\circ$ and 179.8$^\circ$ for the platform orientation and is therewith sufficiently far away from solution I.

The poses calculated from the raw accelerometer values are significantly noisier than the poses calculated with the complementary filtered orientation angles. In fact, the complementary filter's results are at least two times more accurate than the unfiltered results and match the actual manipulator platform's pose quite well. Especially in the $x$-axis, the complementary filter's results are comparatively accurate and do not exceed a position error of $\pm 5$ mm. For the platform orientation, the complementary filter's results show errors mainly between $-10^\circ$ and 5$^\circ$. Only between second 22 and 25 of the experiment, the complementary filter shows strangely big position and orientation errors. The same but with a smaller impact, happens at second 3 of the experiment. In both cases, the calculated

orientation angle of the manipulator platform drifts away by $10°$ (at 3 s) and even $30°$ (at 23 s). The reason for this is probably that, due to the high velocity and the sampling rate, the Arduino Mega looses some measuring information leading to inaccurate linear actuators' orientations.

Figure 13 summarises the position and orientation errors of the raw and the filtered orientation angles in boxplots. Both the poses calculated from the raw accelerometer values and the poses calculated with the complementary filter do not show any offset errors. Due to the high measurement noise, the poses calculated from the raw accelerometer values are very noisy and show huge errors. In total, only 50% of the errors range between $-15.3$ mm and $11.9$ mm for the $x$-axis, between $-23.2$ mm and $22.3$ mm for the $y$-axis and between $-21.9$ mm and $19.3$ mm for the platform orientation. Furthermore, the the proposed formulation often fails to solve Equations (45) and (46) from the raw accelerometer values. Apparently, the root in Equation (47) becomes negative. This can be traced back to the noisy IMUs' measurements under fast motions. Hence, from the raw accelerometer values, the pose cannot be obtained sufficiently accurate. In contrast to that, the results calculated from the complementary filter are significantly more accurate and robust. Here, lower variances of the position and orientation errors can be obtained.

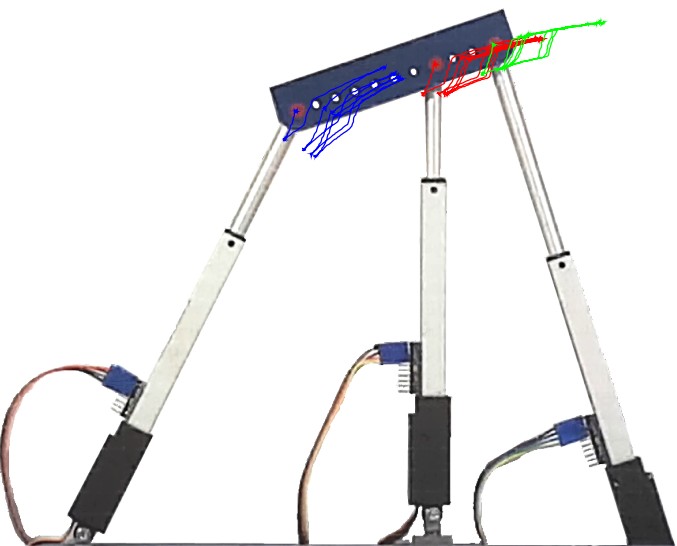

**Figure 11.** Trajectories of the first (blue), second (red) and third (green) manipulator platform joint during the dynamic experiment. The trajectories were recorded by a camera with 30 fps and the joints' positions were analysed using image processing.

For comparison, we additionally used the linear actuators' lengths and the Newton-Raphson algorithm to calculate the manipulator platform's pose iteratively. These results, however, are not calculated on the Arduino Mega due to the required initial estimate in the beginning of the experiment and, more importantly, the significantly longer calculation time. In fact, using the linear actuators' lengths and the Newton-Raphson algorithm is at least ten times slower than using the proposed analytic algorithm together with the filtered orientations. However, even though the actual pose of the manipulator platform was given as an initial pose estimate, for this experiment, the Newton-Raphson algorithm converged to a completely wrong pose in the beginning, that is,

$$\begin{bmatrix} x & y & \gamma \end{bmatrix}^\top = \begin{bmatrix} 172.0 \text{ mm} & -142.4 \text{ mm} & 233.3° \end{bmatrix}^\top , \tag{70}$$

and did not return from there. Hence, the results of the Newton-Raphson algorithm does not match the actual pose at all. Possible reason are the small errors in the linear actuators' lengths and the missing robustness of the Newton-Raphson algorithm.

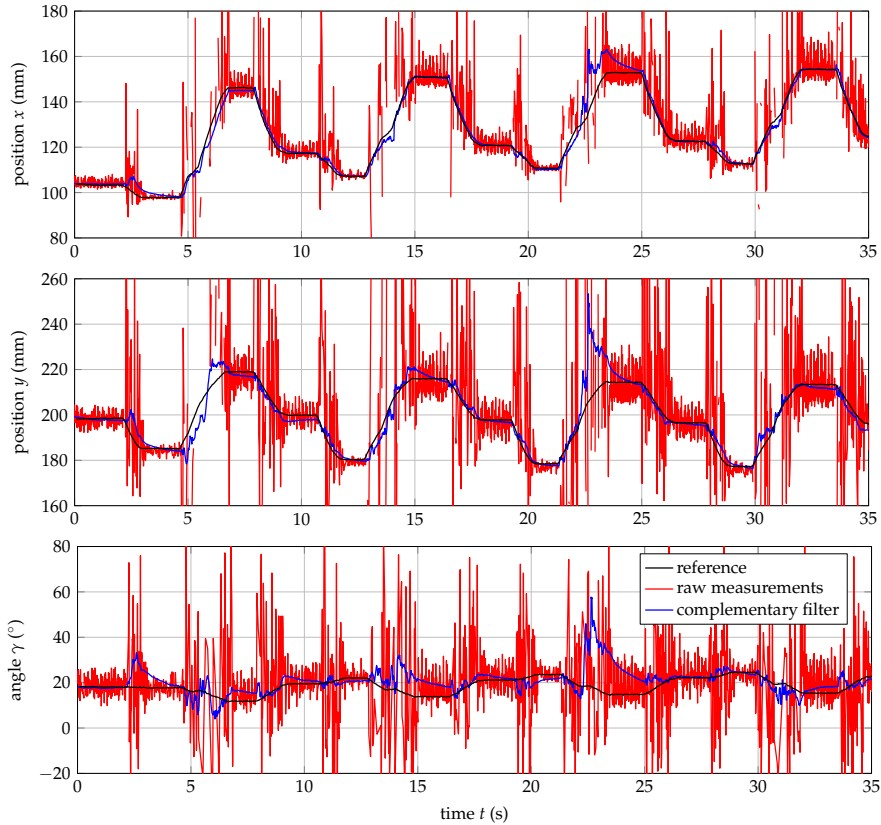

**Figure 12.** Pose of the manipulator platform during the dynamic experiment calculated from the raw (red) and the filtered (blue) linear actuators' orientations: (**a**) *x*-position, (**b**) *y*-position and (**c**) orientation angle $\gamma$. As reference (black), the positions and orientations calculated from the optically analysed manipulator platform joints are used.

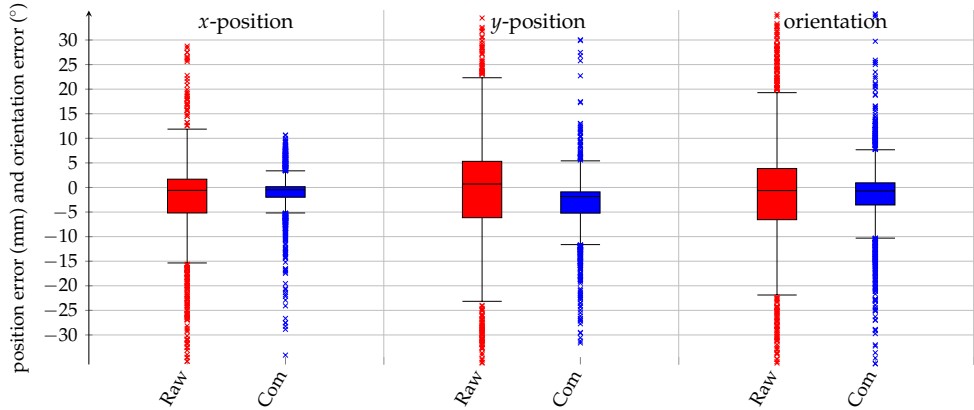

**Figure 13.** Boxplots of the position and orientation errors of the manipulator platform's pose during the experiment calculated with the raw orientation angles (red) and the complementary filtered orientation angles (blue). The box corresponds to the area in which the middle 50% of the errors lie while the whiskers indicate the area in which the middle 99.3% of the errors lie.

In conclusion, the results obtained from the filtered linear actuators' orientation angles together with our formulation proposed in Section 3 are capable of calculating the actual manipulator platform's pose even under dynamic conditions. Comparably small offset errors and variances can be obtained throughout the dynamic experiment. It therewith is significantly more accurate and robust than the

raw orientation angles. Nevertheless, the variances obtained with the complementary filter are still too high for an accurate pose control. However, it outperforms the Newton-Raphson algorithm in terms of accuracy, robustness and computational efficiency.

## 6. Conclusions

In this paper, we first reviewed classical methods for solving the direct kinematics problem of parallel mechanisms and discussed their disadvantages on the example of the general planar 3-RPR parallel mechanism. In order to avoid these disadvantages, we proposed a sensor concept together with an analytical formulation for solving the direct kinematics problem of a general planar 3-RPR parallel mechanism. By measuring the orientations of the linear actuators, provided, for example, by inertial measurement units, the number of possible assembly modes can be reduced down to two when using the linear actuators' orientations instead of their lengths. Finally, we experimentally evaluated the accuracy of our direct kinematics solution under static as well as dynamic conditions by performing experiments on a specially designed prototype.

The static experiments prove that it is possible to calculate the two possible assembly modes of the manipulator platform from the linear actuators' orientations. For the investigated general planar 3-RPR parallel mechanism, the two solutions of the direct kinematics problem are sufficiently far away from each other to distinguish between them. By using the raw accelerometer values to calculate the linear actuators' orientation angles, the variances in the orientation angles are quite high leading to huge variances in the calculated poses of the manipulator platform. The mean results, however, are quite precise. By using a complementary filter instead, where the linear actuators' orientation angles are calculated from the IMUs' accelerometer and gyroscope values, the variances in the orientation angles are significantly smaller (27 to 38 times) leading also to smaller variances in the calculated poses of the manipulator platform. Here, variances between $0.006 \, \text{mm}^2$ and $0.155 \, \text{mm}^2$ for the $x$-axis, between $0.051 \, \text{mm}^2$ and $2.450 \, \text{mm}^2$ for the $y$-axis and between $0.073^{\circ 2}$ and $0.743^{\circ 2}$ can be obtained.

By using the Cramér-Rao lower bound (CRLB) together with the known variances of the linear actuators' orientation angles, it is possible to estimate the variances of the calculated manipulator platform's pose in each axis for every pose in the workspace. For the static measurements, the experimental results match the CRLB's results so that we can conclude that the proposed solution formulation is the optimal estimator with the lowest amplification of measurement variances on the position and orientation variances.

The dynamic experiment also indicates that the raw accelerometer values are too noisy to be used for accurately and robustly calculating the manipulator platform's pose. Throughout the experiment, the results show huge variances. Furthermore, the proposed formulation furthermore fails to solve Equations (45) and (46) that, can also be traced back to the noisy raw measurements. Much more accurate and robust results can be obtained from the filtered orientations angles. The complementary filter shows significantly lower variances of the position and orientation errors and no offset error. Therewith, the proposed analytic algorithm enables to actually calculate the manipulator platform's pose even under dynamic conditions. The risk of confusion between the two assembly modes never existed during the experiments since solution I, provided by Equation (45), always corresponds to the actual pose.

The analytic formulation for calculating the two assembly modes of the manipulator platform from the linear actuators' orientations presented in Section 3 can be further generalized. In fact, in the model of the general planar 3-RPR parallel mechanism, we assumed that the three manipulator platform joints $D$, $E$ and $F$ are aligned. This model is sufficiently general to show that the planar 3-RPR parallel mechanism can have up to six assembly modes. However, it does not correspond to the most general case where no constraints are given for the base and the manipulator platform joints. In future, we will focus on finding an analytic formulation for calculating the assembly modes even for this case.

By obtaining a unique solution of the direct kinematics problem without requiring the linear actuators' lengths, in future, it is possible to actually benefit especially in the control of parallel

mechanisms. Usually, the measured linear actuators' lengths are compared with the target lengths that are provided by inverse kinematics for a given target pose, see Figure 14a. Due to the direct kinematics problem and the existence of singularities in the workspace of parallel mechanisms, we cannot guarantee that the linear actuators' lengths correspond to only one pose and it is possible that the manipulator platform is in (or moves to) a different pose than expected. This problem can be solved by using more suitable coordinates, for example, the linear actuators' orientations. When the direct kinematics problem provides a unique solution or, in this case, the two solutions are far away from each other, we can ensure that the manipulator platform always moves to the target pose. Figure 14b shows a pose control concept where the linear linear actuators' orientation angles $\varphi$ are used. For a given target pose $p_{\text{target}}$, the target orientation angles $\varphi_{\text{target}}$ can be calculated from inverse kinematics. They can be compared with the measured orientation angles $\varphi_{\text{is}}$ and the required deviation of the orientation angles $\Delta\varphi$ can be calculated and given to the controller, for example, a PID controller. The controller then calculates an appropriate output $u$ for the system that, in turn, produces the system output. Using the proposed sensor concept, the system output can be measured, for example, with IMUs mounted on the linear actuators. These measurements are filtered and finally compared with the new target orientation angles. In contrast to usual control concepts where we cannot guarantee that the pose that belongs to the measurements, in general, the linear actuators' lengths, is actually the target pose (indicated by the dashed line in Figure 14a. However, by using the proposed control concept shown in Figure 14b, we actually can. In this context, controllability of the robot is essential. Briot et al. [76] proposed an interesting approach to the analysis of the controllability of parallel mechanisms.

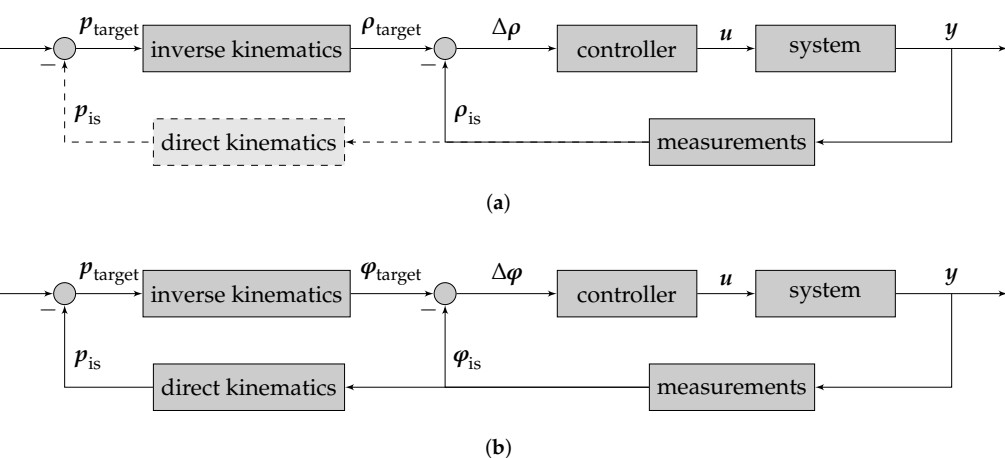

**Figure 14.** Conventional (**a**) and proposed control concept (**b**) for controlling the manipulator platform's pose of a parallel mechanism. The conventional control concept uses the linear actuators' lengths, whereas the proposed control concept uses the linear actuators' orientations. In contrast to the conventional control concept, the proposed control concept can guarantee an analytic solution of the direct kinematics problem.

**Supplementary Materials:** The following are available online at http://www.mdpi.com/2218-6581/8/3/72/s1, Video S1: Demonstration_Video_3RPR. A video including the dynamic experiment, the Matlab code and other information are available online at https://github.com/stefanschulz85/Assembly-Modes-of-a-3-RPR-parallel-Mechanism-when-Using-the-Linear-Actuators-Orientations (doi:10.5281/zenodo.3240459).

**Funding:** This research received no external funding.

**Acknowledgments:** Stefan Schulz would like to thank Arthur Seibel for his valuable suggestions and Aniruidha Nagaraj Vyasamudra for his commitment during his project work. The publication of this article was supported by the Deutsche Forschungsgemeinschaft (DFG, German Research Foundation)—Projektnummer 392323616 and Hamburg University of Technology (TUHH) in the funding programme "Open Access Publishing".

**Conflicts of Interest:** The author declare no conflict of interest.

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
