# Peer review of "Performance Evaluation of a Sensor Concept for Solving the Direct Kinematics Problem of General Planar 3-RPR Parallel Mechanisms by Using Solely the Linear Actuators’ Orientations"

_robotics, doi:10.3390/robotics8030072_

Round 1

Reviewer 1 Report

I have read the paper carefully, I find that it is a paper correctly developed in the introduction, the modeling, and the experimental results. I think it is an interesting paper, especially in the methodology, since other publications have already considered the use of inertial units IMUs in parallel platforms, for example in the case of parallel mechanisms such as Stewart-Gough.

Author Response

Thank you very much for your feedback.

Reviewer 2 Report

This paper introduced additional sensors to solve the direct kinematic problem of the 3-RPR parallel robot. This method is not new and was already presented in the research done by Tancredi in 1995.

As the geometric parameters introduce simplification in the kinematic problem, the author can obtain analytic solutions (See Kong's Thesis), ie 4 solutions. 

The notion of singularity is not introduced in the paper. As the singularity depends on the active joints, we have to take about two problems: control the position of the mobile platform and know its posture. 

The author has to search for information about the aspect to evaluate the maximum singularity free region in the workspace. He will discover the meaning of this problem. 

The author has to take care of the method introduced by Merlet and Gosselin to solve the direct kinematic problem. In ASME JMR, Wenger has investigated the problem when RV-SU is equal to zero.

It's very strange to present one method and later to explain that another is used to solve the direct kinematic problem. We have also to highlight that the method based on the distance can yield two times more solutions because we do not have the order of the points located on the mobile platform in the loop. 

Please present an example where the points located into the base and the mobile platform are not aligned.

I think that we can obtain an analytic solution when we change the sensor location and it is more complex to apply in the control low. In which space will be the regulation of the control?

Author Response

Thank you very much for the detailed review. In the following, we try to address your remarks point-by-point in order to improve the paper. All significant changes are highlighted in blue in the text.

1)      This paper introduced additional sensors to solve the direct kinematic problem of the 3-RPR parallel robot. This method is not new and was already presented in the research done by Tancredi in 1995.

The reviewer is correct by saying that solving the direct kinematics problem with additional sensors is not new. In Section 2.3. (Additional Sensor Solution), we therefore discuss the advantages and disadvantages of known concepts. For example, Tancredi et al. (2015), Stroghton & Arai (1991), Merlet (1993) and Parenti & Di Gregorio (2000) suggested using rotary sensors as additional sensors. We have added these and other papers in order to further elaborate on this point. However, none of the existing concepts completely renounces the lengths of the linear actuators and solely uses the orientations for solving the direct kinematics problem. They always use the linear actuators’ lengths in addition to other sensor information, for example, the rotary sensor data. This is the main difference to our paper. In our work, solely the linear actuators’ orientations are used (measured by IMUs/rotary sensors) for solving the direct kinematics problem. In order to make the difference of our approach to existing methods clearer, we added an additional paragraph in the manuscript.

2)      As the geometric parameters introduce simplification in the kinematic problem, the author can obtain analytic solutions (See Kong's Thesis), ie 4 solutions.

We added a paragraph in the manuscript where we addressed the work by Kong and Gosselin (2001).

3)      The notion of singularity is not introduced in the paper. As the singularity depends on the active joints, we have to take about two problems: control the position of the mobile platform and know its posture.

The reviewer is correct. We have introduced an additional paragraph in the manuscript and cited relevant papers (with respect to the 3-RPR parallel mechanism) to clarify the notion of singularity.

4)      The author has to search for information about the aspect to evaluate the maximum singularity free region in the workspace. He will discover the meaning of this problem.

We agree with the reviewer. However, finding singularities or the maximum singularity free region is not the scope of this paper. In future research, we will focus on singularities, singularity avoidance, and their impact on control.

5)      The author has to take care of the method introduced by Merlet and Gosselin to solve the direct kinematic problem. In ASME JMR, Wenger has investigated the problem when RV-SU is equal to zero.

Thank you for mentioning these papers. We are aware of them but we missed mentioning them in the manuscript. We therefore added additional paragraphs in the manuscript. In fact, we have addressed relevant papers in the context of the 3-RPR parallel mechanism.

6)      It's very strange to present one method and later to explain that another is used to solve the direct kinematic problem.

Thank you for your remark. We presented, i.e., reviewed, classical methods for solving the direct kinematics problem and visualized possible problems and associated disadvantages on the example of the 3-RPR parallel mechanism. This is done to motivate our sensor concept and the solution formulation for solving the direct kinematics problem. In order to clarify this, we have introduced an additional paragraph in the manuscript.

7)      We have also to highlight that the method based on the distance can yield two times more solutions because we do not have the order of the points located on the mobile platform in the loop.

Thank you very much for your remark. We have included this problem in the introduction section of the manuscript.

8)      Please present an example where the points located into the base and the mobile platform are not aligned

This is an interesting point because this would be the most general case of the 3-RPR parallel mechanism. We are working on this problem but we have not solved it yet. We added a corresponding paragraph in the conclusion section of the manuscript.

9)      I think that we can obtain an analytic solution when we change the sensor location and it is more complex to apply in the control law. In which space will be the regulation of the control?

Changing the sensor location can provide even a unique solution as discussed, for example, in previous work (see below) where we measured the orientation of the manipulator platform and two of the linear actuators’ orientations. The presented control strategy in the manuscript allows regulations in the joint space (i.e., controlling the linear actuators’ orientations) as they provide an analytic solution of the direct kinematics problem. The implementation of this approach in the control law is a topic we are currently working on.

Schulz, S.; Seibel, A.; Schlattmann, J.: Closed-form solution for the direct kinematics problem of the planar 3-RPR parallel mechanism. In: Proceedings of the IEEE International Conference on Robotics and Automation (ICRA), pp. 968-973, Brisbane, QLD, Australia, 2018.

Schulz, S.; Seibel, A.; Schlattmann, J.: Assembly Modes of General Planar 3-RPR Parallel Mechanisms when Using the Linear Actuators' Orientations. In Uhl,T. (Ed.): Advances in Mechanism and Machine Science. IFToMM WC 2019. Mechanisms and Machine Science, Vol 73., Springer, Cham  pp. 279-288, 2019.

Round 2

Reviewer 2 Report

The new version does not change the geometry of the robot. The results do not allow to test the method with a 3-RPR robot with 6 solutions to the direct kinematic model .

Please read

Briot, S., Martinet, P., & Rosenzveig, V. (2015). The hidden robot: an efficient concept contributing to the analysis of the controllability of parallel robots in advanced visual servoing techniques. IEEE Transactions on Robotics, 31(6), 1337-1352.

It is absolutely necessary to repeat the experimental part by taking up the general case. Currently, the problem is too simple.

Author Response

Thank you very much for the review. We repeated the experiments with the general case 3-RPR mechanism with six possible solutions for the direct kinematics problem that is introduced and discussed in the introduction section. The experiments show the validity of our proposed method even for this mechanism.

As the experimental results slightly changed, several changes were made in the experimental part of the paper. All significant changes are highlighted in green in the text.